# Short Chain Fatty Acid Metabolism in Relation to Gut Microbiota and Genetic Variability

**DOI:** 10.3390/nu14245361

**Published:** 2022-12-16

**Authors:** Guilherme Ramos Meyers, Hanen Samouda, Torsten Bohn

**Affiliations:** 1Nutrition and Health Research Group, Department of Precision Health, Luxembourg Institute of Health, 1 A-B, Rue Thomas Edison, 1445 Strassen, Luxembourg; 2Doctoral School in Science and Engineering, University of Luxembourg, 2, Avenue de l’Université, 4365 Esch-sur-Alzette, Luxembourg

**Keywords:** nutrigenetics, nutrigenomics, dietary fibre, short chain fatty acids, microbiome, synergies, sustainable development, holobiont, SNPs, translational research

## Abstract

It is widely accepted that the gut microbiota plays a significant role in modulating inflammatory and immune responses of their host. In recent years, the host-microbiota interface has gained relevance in understanding the development of many non-communicable chronic conditions, including cardiovascular disease, cancer, autoimmunity and neurodegeneration. Importantly, dietary fibre (DF) and associated compounds digested by the microbiota and their resulting metabolites, especially short-chain fatty acids (SCFA), were significantly associated with health beneficial effects, such as via proposed anti-inflammatory mechanisms. However, SCFA metabolic pathways are not fully understood. Major steps include production of SCFA by microbiota, uptake in the colonic epithelium, first-pass effects at the liver, followed by biodistribution and metabolism at the host’s cellular level. As dietary patterns do not affect all individuals equally, the host genetic makeup may play a role in the metabolic fate of these metabolites, in addition to other factors that might influence the microbiota, such as age, birth through caesarean, medication intake, alcohol and tobacco consumption, pathogen exposure and physical activity. In this article, we review the metabolic pathways of DF, from intake to the intracellular metabolism of fibre-derived products, and identify possible sources of inter-individual variability related to genetic variation. Such variability may be indicative of the phenotypic flexibility in response to diet, and may be predictive of long-term adaptations to dietary factors, including maladaptation and tissue damage, which may develop into disease in individuals with specific predispositions, thus allowing for a better prediction of potential health effects following personalized intervention with DF.

## 1. Introduction

The human organism is composed of eukaryotic cells, as well as of an assembly of microbes collectively termed the microbiota, including archaea, bacteria, fungi and eukaryota. These may outnumber human cells, although a 1:1 ratio seems more likely, according to more recent estimates [1]. Regardless of the quantity of genes within individual microbial cells, the microbiome (the whole genome of the microbiota) encompasses over 1000 microbial species. Thus, the microbiome complements the human genome in functionality, such as enhancing digestion or protecting from pathogenic invasion [2,3]. The largest fraction of microbiota is found in the colon, and is termed, together with a smaller fraction residing in the stomach and small intestine, the gut microbiota [4]. Indeed, evolutionary biology proposes an analogous eukaryon-mitochondrion symbiosis that occurred between multicellular eukaryotes and prokaryotes millions of years ago, the so-called holobiont theory [2].

Evidence is mounting that the gut microbiota (GM) plays a fundamental role in regulating metabolic, immune and endocrine functions, as well as priming the immune response against pathogens. Indeed, GM alterations such as total abundance of or ratios between different species or families have been associated with many different health issues [5], specifically those of non-communicable chronic diseases (NCDs) such as obesity [6], cardiovascular disease and atherosclerosis [7], type 2 diabetes (T2D) [6,8], autoimmune disorders such as rheumatoid arthritis [9], ageing conditions, e.g., osteoporosis and sarcopenia [10], neurodegenerative diseases [11,12] including Parkinson’s [13] and Alzheimer’s disease [14], as well as several types of cancer [15,16]. In addition to GM changes, the majority of these conditions is characterized by a low-grade chronic inflammation [17,18,19,20], concurring with increased levels of oxidative stress [21,22].

Research has highlighted the significant and strong relationship between dietary patterns and the development of NCDs, such as CVD, depression, cognitive decline, multiple sclerosis, Parkinson’s disease, osteoarthritis and gastrointestinal conditions such as irritable bowel syndrome (IBD) [7,23,24,25,26,27,28,29,30,31,32,33,34], with much attention being dedicated to dietary fibre (DF) [35]. Overall, a higher DF intake has been associated with reduced all-cause mortality, e.g., in the Asian population [27], and hypotheses on its role as a health protective factor have been existing for several decades [36]. Studies have demonstrated improved health outcomes with higher fiber intake in conditions ranging from *C. difficile* infection [37] to paediatric kidney disease [38], showing its wide applicability in health maintenance. Regrettably, in most countries, it appears that DF intake has been on the decline. In Japan, where data are available since the 1950s, a 30% drop in DF intake was observed between the 1950s and 1970s, and then stabilized—though this may be subject to change, as younger generations report far less DF intake than their elders [39]. A review by the *Nutrition Society* [40], as assessed by national surveys in the UK, revealed a DF intake of approximately 14.8 g/d in adults, men and women, in 1999 [41], and about 13.6 g/d in 2009–2012 [42]. In the USA, DF intakes remained stable from 1999 to 2008, but well below recommendations, at around 15 g/d [43]. Concurrently, the highest consumption of DF in Europe was found in Germany (25 g/d for males and 23 g/d for females), based on a telephone-survey performed in 2005–2006 [40], being in line with EFSA recommendations.

DF may be at the centre of the symbiotic relationship between the GM and the human host [35,44,45,46,47,48,49]. DF is not absorbed or broken down to a significant degree by human digestive enzymes, and can, at least in part, be used as an energy substrate by the GM. Depending on the nature of DF, it is predominantly metabolized into short chain fatty acids (SCFA), including butyrate, acetate, and propionate [44]. Butyrate, acetate and propionate cross the enterocyte layer and are absorbed, while lactate and succinate appear to be intermediate products of DF fermentation [50]. Immediately, butyrate acts as the main energy source for colonocytes and controls maturation of mucosa associated lymphoid tissue (MALT) [51,52,53,54,55,56,57,58,59,60,61,62,63], characterized by a high presence of immune cells such as macrophages, B and T cells and that plays an important role in antigen sensing. Only a fraction of the produced SCFA enter the host’s systemic circulation, with acetate corresponding to around 75% of total peripheral SCFA [64,65]. However, these values have shown a high degree of inter-individual variation, as well as intra-individual variation such as dose–response, time-course and circadian variance [66]. SCFA may act as pleiotropic immunomodulators, i.e., having different functions in different tissues [35,51,67]. SCFA appear to be strong influencers of immune regulation, as seen in studies regarding asthma and atopy in infants, as well as in mice models [68,69,70,71,72], or gastrointestinal health in adults [35,48,51,52,73,74,75]. As described in the following chapters, SCFA production and concentrations were associated with disease risk. In addition to SCFA, DF acts as a vehicle for antioxidants in the upper gastrointestinal tract [76,77], as it is associated with a large number of phenolic compounds [44,78,79,80,81] and other secondary plant metabolites such as carotenoids [44,80]. Especially phenolic compounds may likewise be turned into bioactive metabolites by the GM [77,82], and synergies between these food derived compounds may exist, further highlighting their importance [83,84,85,86,87].

Apart from drugs, age, delivery method, medication intake, alcohol and tobacco consumption, pathogen exposure, besides diet in general, and dietary secondary plant metabolites in particular, are known to be significant modulators of the GM [44,88,89,90,91,92,93,94]. Dietary antioxidants can alter GM composition and thus its products [95]. However, the genetic background also modulates bacterial colonization [3,96]. In particular, genetic variants such as single nucleotide polymorphisms (SNPs), may further explain some of the inter-personal variability observed following fibre intake, such as circulating levels of SCFA [53,72,97,98]. Variations in genes such as *GPR41, GPR43* or *GPR109A* (G-protein coupled receptors for SCFA) [99] could have substantial impact on the immunometabolism of certain tissues in particular, and the organism in general. Furthermore, transporter genes of the *SLC16A* family (monocarbohydrate transporters), effector genes such as *MUC2* (for mucus layer production in the colon) or regulatory genes such as *NRF2* (regulating the expression of proteins involved in the bodies’ antioxidant defence mechanism such as superoxide dismutase (SOD)), could have important downstream effects on health outcomes (Figure 1) due to impaired absorption of SCFA or by impacting their functions intracellularly [100].

In this review, we aim to relate the relationship between the metabolism of short chain fatty acids (SCFA) and the host-genetic background. In particular, we will investigate the genetics associated with differences in terms of SCFA production at the GM level and its metabolism at the host level and relationship to health.

## 2. Dietary Fibre and Short Chain Fatty Acids

### 2.1. Dietary Fibre (DF)

Westernized types of diet are characterized by a relatively low intake in DF, despite attempts to increase its intake since the 1970s. Most European countries have established recommendations on daily intake for DF, e.g., 25–35 g for adults. Concretely, 25–32 g/d for adult women and 30–35 g/d for adult men, while recommendations for children and older adults depend on age, being approximately 3–4 g/MJ [40]. The Physicians Committee for Responsible Medicine (PCRM) of the US recommends even a considerably higher intake of 40 g/d for an optimal health [101].

The European Food Security Authority (EFSA) has recommended an adequate intake (AI) of 25 g/d for DF, mostly based on its association with improved bowel function (as per defecation frequency and transit time), and the reduction of gastro-intestinal symptoms such as constipation [102]. DF refers to total fibre occurring naturally in foods such as fruits, vegetables, pulses and cereal grains [40,102]. Grain products are at present the largest source for DF intake worldwide, providing approx. 32% of total dietary fibre intake in the USA and 48% in the Netherlands. Other sources vary widely in European countries, e.g., vegetables (12–21%), potatoes (6–19%) and fruits (8–23%) [40]. Lack of DF intake has been emphasized as one of the major dietary factors associated with the increased incidence of NCDs [103,104,105,106]. A recent systematic review and meta-analysis suggested that high DF consumption was associated with a 15–30% decrease in cardiovascular-related mortality, T2D and colorectal cancer, when compared with low-fibre consumption [107]. Concurring dietary factors such as increased sugar consumption, increased saturated fat consumption and low nutrient density, among others, and their possible relationship to metabolic and neurophysiological disorders, may be present and are expected to play a role [40,108]. However, as human lifespan has expanded during the past decades [109,110], we expect to face an increase of NCDs, as these are rather associated with age-related chronic inflammation (i.e., inflammageing [18]). Therefore, it is paramount to fully understand the pathophysiology of NCDs, and how to counteract them with affordable and efficient strategies, including improved dietary patterns and healthy food items [18,110,111,112,113,114,115,116,117]. In this respect, fiber intake could be increased both within a low-fat diet a low-carbohydrate diet. A randomized controlled trial aiming at weight reduction over a period of 12 months assessed sources of DF in a balanced low-fat diet vs. a balanced low-carbohydrate diet. A large proportion of DF for both diets was from non-starchy vegetables. While the low-fat group mainly increased DF intake from whole grains and fruits, the low-carbohydrate one obtained DF rather from vegetables and plant protein sources. This was further reflected in gut microbiota alterations throughout the intervention, and such dietary adaptations may constitute an important factor for precision nutrition [118].

A variety of definitions has been proposed to classify DF; most were dependent on the methods used to extract DF. This led to difficulties in defining the term, as most non-starch polysaccharides (NSP) were retrieved by such methods, which often did not include resistant (i.e., non-digestible) starches (RS). DF can further be categorized based on its solubility, fermentability or viscosity, which often caused distinctions within the group. While soluble fibres can be fermented to different degrees, and are the main substrate for colonic fermenters (e.g., β-glucans), insoluble fibres mainly serve a stool bulking function (e.g., cellulose). Both types of DF have beneficial health properties, and as such, the dichotomy of soluble-insoluble may no longer play a main role in terms of public health.

To date, definitions have reached a certain consensus [119,120]. DF is composed of carbohydrate polymers with three or more monomeric units (MU), which are neither hydrolysed by human digestive enzymes nor absorbed in the human intestine, and include NSPs from fruits, vegetables, grains and tubers, whether intrinsic or extracted, either chemically, enzymatically, or in physically modified forms. Polymers with more than 10 MU, e.g., cellulose, hemicelluloses, pectins, hydrocolloids (i.e., gums, β-glucans, mucilages); resistant oligosaccharides, e.g., fructo-oligosaccharides (FOS), galacto-saccharides (GOS) with 3–9 MU; and RS with 10 or more MU [40] are included. Furthermore, some constituents produced by micro-organisms (e.g., xanthan) and polysaccharide constituents of crustaceans and fungi (e.g., chitin, chitosan, chondroitin sulphate), are resistant to digestion and are included in the DF definition, according to some national agencies [40]. Furthermore, it has been proposed that proteins resistant to digestion exist, and may reproduce similar effects as DF, namely improved bowel function and improved immunity [121,122,123], but these are typically not included in the DF definition.

Thus, DF is any polymeric carbohydrate not digested in the small intestine. DF generally also includes substances associated with, or linked to plant cell walls, but that are not carbohydrates, such as lignin or polyphenols. Often, these distinctions are not reported in food tables, where only the sum of DF is given. In 2002, the French Agency for Food Security (ANSES), included in its definition all of the above polymeric carbohydrates (MU ≥ 3) as DF, while excluding animal-based sources and lactulose, a non-absorbable sugar, to prevent its incorporation into foods (as it is a strong laxative) as a fibre source [124].

Within this manuscript, DF is considered as any polymeric compound, which is not digestible by human enzymes and which mainly travels through the gut to reach the colonic milieu, where it is either fermented by colonic bacteria (i.e., broadly, soluble fibres) into smaller molecules such SCFA, or can act as a bulking agent during stool production (i.e., generally insoluble fibres). This broader definition would thus also include non-carbohydrate compounds such as lignin and resistant proteins, as well as compounds associated with plant-based carbohydrates, such as polyphenols. These compounds may also be substrates for bacteria, such as *Akkermansia*, *Lactobacillus* and *Bifidobacterium*, which produce metabolites such as SCFA, which in turn induce various beneficial effects on the host, including reduction in: appetite, insulin resistance, lipid accumulation, and inflammation [100]. However, the effects of phytochemicals are likely to vary according to the composition of the gut microbiota and host genetic polymorphisms, which affect absorption, detoxification, and overall bioactivities [125]. One such example is equol, produced form the isoflavone daidzein, which may bind to β-oestrogen receptors, and has been associated with the incidence of various types of hormone-associated cancers [126]. This is in line with the definition proposed by Jones [127], and may overcome the matter of “functionality” often discussed regarding DF, as previously pointed out [128].

Fibre fermentation relies on its chemical and physical structure, as well as the composition of the colonic microflora. Digestion of DF by the GM may vary or fluctuate depending on which fibres are consumed, and thus the amounts of SCFA produced too. For example, lignin and cellulose are rather lost through the stool, being insoluble bulking fibres; polysaccharides from extremely hard plant tissue areas are also less well digestible because physical encrustation and chemical bonding to lignin can occur [46]. Oligosaccharides, RS and pectins are the DF compounds thought to contribute the most to SCFA production in the colon [35].

### 2.2. Short Chain Fatty Acids (SCFA)

Recent studies on DF, GM and probiotics have emphasized the role of SCFA. Indeed, SCFA may be a good example of microbiota-derived modulator molecules, i.e., a nutrient that can modulate the host, acting as communicating molecules between the GM and the host [66]. Provided that SCFA metabolism may have a broad range of implications for human health, many studies are being conducted to understand their effects (Table 1). Sakata [66] recently pointed out relevant pitfalls in the study of these molecules. SCFA are defined as volatile fatty acids with a skeleton of six or less carbons in straight (C1, formate; C2, acetate; C3, propionate; C4, butyrate; C5, valerate; C6, caproate), or branched-chain conformation (C4, isobutyrate; C5, isovalerate and 2-methyl-butanoate). Acetate (C2), propionate (C3) and butyrate (C4) amount for 90–95% of total GM SCFA output and are derived from carbohydrate fermentation [129,130]. Until recently, caproate [131] and valerate [132] were considered dietary food components. However, recent studies have demonstrated that these may also be GM products, with caproate being significantly increased in faecal samples of volunteers with severe obesity (BMI ≥ 40) [131].

Branched-chain SCFA (BCFA), mainly isobutyrate, isovalerate and 2-methylbutanoate, contribute to as much as 5% of total SCFA production, and arise from the metabolism of the amino acids valine, leucine, and isoleucine, respectively [129,131]. BCFA levels in faecal samples show an inverse correlation with fibre consumption, especially insoluble fibre [131,133]. BCFA levels in stool have also been related to depression [32,34] and other psychiatric conditions [134], possibly through vagal afferent nerve signalling [135]. Furthermore, BCFA were found to be increased in subjects with hypercholesterolemia compared to normocholesterolemic individuals, with isobutyrate being associated with worse serum lipid profiles [136]. It is likely that such elevated BCFA correspond to high protein intake, such as from meat-based diet and a reduced DF intake, which are likewise associated with negative health outcomes and ageing related health complications [131].

Recently, products of DF fermentation have been termed post-biotics [137]. In human adults, the principal products of DF fermentation are SCFA together with certain gases (CO_2_, CH_4_, and H_2_), which may be taken up by the host, or excreted [50]. Production of SCFA in the colon accompanies the bacterial consumption of ammonia, H_2_S and BCFA in the synthesis of protein components for the microbial cell. Therefore, the reduction of these metabolites may also be, at least in part, responsible for the health benefits attributed to SCFA [66], as in addition to BCFA also ammonia [138] has been related to negative health outcomes such as neurotoxicity and hepatotoxicity, as well as increased intestinal permeability, loss of tight junction proteins and increase in pro-inflammatory cytokines as found in animal studies [139]. H_2_S, hydrogen disulphide, may be associated with neurological, cardiovascular and metabolic diseases, when abnormally produced [140].

In this review, SCFA describes, “saturated unbranched alkyl group monocarboxylic acids of 2 to 4 carbon atoms”, referring to acetate (C2), propionate (C3) and butyrate (C4). We will briefly mention valerate (C5) and caproate (C6). It excludes BCFA, as well as succinate and lactate, which are rather intermediate products in GM metabolism, and therefore their concentrations in human serum are related rather to human metabolism, and not influenced considerably by GM or intestinal absorption.

**Table 1 nutrients-14-05361-t001:** Identified effects of SCFA in human interventional, observational, and animal studies.

SCFA	Study (Sample)	Study Design	Tissues Investigated	End-Point Measured	Observed Effects	Reference
Human interventional studies						
C2	H (*n* =32)	Case-control	Peripheral blood	Immunopharmacological effects of Ringer’s acetate	Increased polyclonal antibody production and NK cell activity in healthy and cancer subjects	[141]
C3	H (*n* = 6)	Cross-over	Serum and stool	Blood lipids and glucose, stool bulk and microbiota	C3 supplementation lowers blood glucose. Lipid changes not significant; increase in stool bulk and Bifidobacteria after 1 week intervention	[142]
C4	H (*n* = 16)	Cross-over	Sigmoid colon biopsies and plasma	Oxidative stress markers in colon; CRP, calprotectin; histological inflammation	Rectal administration significantly reduced uric acid and increased GSH. No significant changes in other parameters	[143]
Human Observational studies						
C2-C6	H (*n* = 232)	Observation	Stool	Levels of faecal SCFA and BCFA association with BMI and age	BCFA strongly correlated with age, but not with BMI; BCFA negatively associated with fibre consumption; BMI ≥ 40 showed significantly higher production of SCFA, total BCFA, isobutyrate, isovalerate and caproate SCFA production decreases with age	[131]
Animal (interventional) studies						
C2, C3	M (*n* = 15)	Knock-out	Adipose tissue	Effects of GPCR43 activation	Reduction of lipolysis, reduced plasma free fatty acids levels without flushing associated with GPCR109A	[144]
C2, C3	M (*n* = 12)	Case-control	Adipose, gut, vascular and mesenchymal tissues	GPCR41 and GPCR43 mRNA expression	GPCR43 activation promoted adipose differentiation via PPARγ2. No effects on GPCR41	[145]
C2, C3, C4	S (*n* = 10)	Case-control	Portal and peripheral blood, liver	Food intake following SCFA infusions	Dose-dependent depression in food intake, explained by C3 content in portal vein, which resolved with portal plexus denervation	[146]
C3	R (*n* = 20) P (*n* = 12, 60)	Case-control	Portal blood and liver	Cholesterol synthesis and distribution	Supplemented C3 likely absorbed in the stomach Dose-dependent hypocholesterolemic effect likely due to redistribution of cholesterol from plasma to liver, as opposed to synthesis inhibition	[147,148]
C3	R (*n* = 74, 114)	Case-control	Brain, intracerebral ventricles	Behavioural, electrophysiological, neuropathological, and biochemical effects	C3 intraventricular infusion impaired social behaviours, similar to those seen in human ASD; induced neuroinflammation and oxidative stress; Alteration of brain phospholipid and acylcarnitine1 profiles	[149,150]
C4	R (*n* = 22)	Case-control	Duodenum, jejunum, cecum and distal colon	PYY and proglucagon gene expression in gut epithelial cells	Up-regulation of local peptide YY and proglucagon expression via colonocyte sensing following a RS diet in vivo, proved by in vitro incubation with butyrate	[151]
C4	M (*n* = 16–20)	Case-control	Whole-body autopsy	Insulin sensitivity and energy metabolism, mitochondrial function	C4 supplementation prevented diet-induced insulin resistance and reduced adiposity in high-fat model, without reducing food intake. Attributed to enhanced mitochondrial activity and thermogenesis	[152]
In Vitro Studies						
C2-C6	M (*n* = 18)	N/A	mouse adipocyte cell line and adipose primary culture	Leptin expression	C2-C6 stimulate leptin expression via GPCR41 Acute administration of C3 increased leptin levels	[153]
C2, C4	R, B	N/A	Anterior pituitary, fat and liver aspirates	Leptin and leptin-receptor protein expression	C2 and C4 enhanced leptin expression in bovine pituitary and fat cells, however C4 inhibited leptin expression in rat anterior pituitary cells; while C4 suppressed leptin receptor expression in both rat and bovine pituitaries; probable species specific nutrient sensing	[154]
C2, C3, C4	R, H	N/A	Colonic stimulation	Effects on colon functions, inc. motility	C3 and C4 induced phasic and tonic contractions of circular muscle via GPCR41 and GPCR43 in mucosae, C2 did not	[155]
C2, C3, C4	M (*n*= 4) H (*n*= 3)	N/A	Human blood samples, colon cultures (colo320DM) and mice with colitis	Anti-inflammatory properties of SCFA	All SCFA decreased neutrophil TNF-α release without affecting IL-8; all decreased IL-6 release; all inhibited NF-κB activity in colon cells; C4 > C3 > C2	[156]
C3	H (*n* = 5–9)	N/A	Human umbilical vein endothelial cells (HUVEC)	Expression of endothelial leukocyte adhesion molecules and leukocyte recruitment by cytokine-stimulation	Significant inhibition of TNF-α and NF-κB, reducing expression of VCAM-1 and ICAM-1 in a time- and dose-dependent manner; significantly increased PPARα expression	[157]
C3	H (*n* = 28)	N/A	Omental and subcutaneous adipose tissue	Adipokine expression	Significant leptin induction and secretion; no effect on adiponectin; Reduction of resistin mRNA expression	[158]
C3	R, H (*n* = 1)	N/A	Human blood and rat mesenteric lymph nodes	T and B lymphocyte proliferation and metabolism	Inhibition of lipid synthesis as a possible mechanism leading to reduction of lymphocyte proliferation	[159]
C3	R (*n* = 9)	N/A	Isolated hepatocytes	Hepatic lipidogenesis	Inhibits hepatic cholesterol and fatty acid synthesis in a dose-dependent manner, possibly by competition with C2	[160]

ASD, autism spectrum disorder; B, bovine; H, human,; M, mice; P, pigs; R, rat; S, sheep; C2, acetate; C3, propionate; C4, butyrate; C5, valerate; C6, caproate; HUVEC, human umbilical vein endothelial cells; TNF-α, tumour necrosis factor alpha; VCAM-1, vascular cell adhesion molecule-1; ICAM-1, intracellular adhesion molecule-1; RS, resistant starch; GSH, glutathione peroxidase; PYY, peptide YY; SCFA, short chain fatty acids; BCFA, branched-chain fatty acids; BMI, body mass index; GPCR, G-protein coupled receptor; TNF-α, tumour necrosis factor alpha; NF-κB, nuclear factor kappa-light-chain-enhancer of activated B cells N/A, not applicable.

## 3. Inter-Individual Variability

DF intake does not appear to produce equal results in all individuals [161]. Indeed, this is observed for most nutritional components, and a limitation of conventional nutritional studies [162]. Both host-related factors, but also food matrix related aspects, may play a role. As for vitamins, DF-metabolite bioavailability may be influenced by the SLAMENGHI factors (i.e., molecular species, linkage, amount, matrix, effectors of absorption, nutrition status, genetics, host-related factors, and the interaction of these) [163]. An additional problem regarding DF, at least when comparing results across studies is the variability of DF definitions. A systems analysis approach, currently recommended in clinical oncology [164], and taking its place in other biomedical disciplines [162,165,166,167], may be required to better understand factors explaining inter-individual variability of DF associated effects.

Foremost, identification of different levels of variability in human populations is required. Until recently, the basal level of variation of the effects of DF consumption on health outcomes had not been significantly studied, i.e., the different GM found in humans. One may refer to this level as the enterotype [168,169]. A matter of debate among the scientific community, the enterotype level, attempts to stratify populations according to GM prevalence and abundance. Indeed, depending on each individual’s GM composition, the impact of DF intake on its metabolism and related outcomes may change significantly [170]. DF is associated with reduced transit times [35,171], increased frequency of bowel movements [172,173] and overall improvement of bowel health [35,75,129,174]. However, in individuals with typically low DF intake to whom a rich DF diet has been imposed, adverse effects may arise, such as bloating and intestinal discomfort [175]. Thus, depending on the host’s pre-existent microbiota, the degree of DF fermentation, and therefore SCFA production and their uptake will vary. In the following sub-chapters, we will summarize the three strata of inter-individual variability (enterotypes, genotypes and phenotypes), while focusing on adults free from disease. We will leave aside known differences found in this topic between geographical regions, i.e., countries and continents [176].

### 3.1. Enterotypes—SCFA Production and Relation to Disease

Gut bacterial composition is determined by a myriad of factors. On one hand, host factors (age, genetics, digestive secretion and physiology, immune status, use of medication, e.g., metformin or antibiotics) [177] and environment (geography, diet, environmental pollutants) do play a role [178,179]. On the other hand, microbial factors (substrate competition, metabolic cooperation or species antagonism), as well as microbial environment (local pH, redox potential, quorum sensing) drive the GM composition. *Firmicutes, Bacteroidetes, Proteobacteria, Verrucobacteria, Actinobacteria* and *Fusobacteria* are found ubiquitously in the GM, with 99% of the species falling into the phyla *Bacteroidetes* or *Firmicutes* in adulthood. These two phyla represent 70% of the total GM [180]. In the first years of life, the GM is mainly composed of Proteobacteria and Actinobacteria, although this depends largely on delivery mode and feeding mode in infancy. A recent study studying over 2700 families found that around 6.6% of taxa are heritable (especially Proteobacteria, *A. muciniphila*, *Bifidobacterium longum*) consistent with previous twin studies [181], while around 48.6% of taxa is significantly explained by cohabitation.

SCFA producing bacteria are known as DF fermenters (Table 2). The effect of DF interventions on the GM of healthy adults has been reviewed elsewhere [182], as well as SCFA production by the microbiota [129] (Figure 2). Whereas a meta-analysis revealed considerable heterogeneity in results, significant relationships between specific DF interventions, GM communities and SCFA production, could be made. Particularly, glycans and GOS led to significantly greater abundance of both *Bifidobacterium* spp. and *Lactobacillus* spp. compared with placebo and low fibre diet comparators. Faecal butyrate concentration was significantly increased when compared to placebo/low-fibre regimens, although heterogeneously across studies [182]. In short, acetate and propionate were mainly produced by *Bacteroidetes*, whereas the *Firmicutes* phylum tended to produce butyrate [183]. This can have repercussions on the inflammatory state of the host (see following chapter), as especially butyrate has been related to anti-inflammatory properties.

Because bacteria tend to organize based on interspecies metabolic relationships, the notion of enterotypes has been proposed [168]. Enterotypes do not occur as discrete clusters, but instead in gradients, with groups tending towards preferred genus level composition [184]. The abundance distribution of different microbial taxa is thus complex. Nevertheless, networks of co-occurring microbes have been described, whose regulator (driver) taxon could be identified, i.e., a taxon that best correlates among bacterial group tendencies [168]. These are:Enterotype 1, or ET-B, presenting *Bacteroides* as the taxon driver;Enterotype 2, or ET-P has *Prevotella* genus as common denominator—abundance of *Prevotella* is inversely correlated with *Bacteroides*;Enterotype 3, or ET-F is characterized by an abundance of *Firmicutes*, namely *Ruminococcus*.

ET-F displays a positive association with *Akkermansia* spp., a known mucin-degrader, and with *Methanobrevibacter smithii*, the most abundant and prevalent methane-producer in the human gut [185], which in turn are negatively associated with *Prevotella* [168].

In a recent review, growth performance and diarrheal states in pigs and *Prevotella* spp. abundance were investigated [186]. In pigs, ET-P was positively associated with luminal IgA secretion as well as increased body weight. Compared to ET-B, ET-P was associated with 2–3 times more propionate production, following a reduction in butyrate. ET-P was found to associate with chronic inflammation and colitis in pigs, possibly due to reduced IL-18 production. This finding contrasts with mechanistic mice models, where *Prevotella* (an acetate-producer) was found to cross-feed *Roseburia* and *Faecalibacterium* spp. (butyrate-producers), regulating the host’s immunity via increased IL-10 production and receptor-dependent repression of claudin-2, important for tight junction integrity [187]. Furthermore, *Prevotella copri* was found to modulate *Listeria monocytogenes* infection in piglets. In humans, high *Prevotella* abundance was associated with autism spectrum disorders (ASD), rheumatoid arthritis and HIV in individual studies. However, following a meta-analysis, Duvallet et al. have found no association between *Prevotella*, ASD and rheumatoid arthritis. In the case of HIV, the association with *Prevotella* was likely due demographic factors unrelated to disease [188]. Gacesa et al. also found that in humans, seemingly unrelated diseases did share a common microbiome signature independently of comorbidities [189]. This study identified *Prevotella copri* as driving two distinct clusters, where *P. copri* abundance positively associated with general health. In this cohort, microbial disease signatures were consistently related to increases in *Anaerotruncus*, *Ruminococcus*, *Bacteroides*, *Holdemania*, *Flavonifractor*, *Eggerthella* and *Clostridium* species and decreases in *Faecalibacterium*, *Bifidobacterium*, *Butyrivibrio*, *Subdoligranulum*, *Oxalobacter*, *Eubacterium* and *Roseburia*. The differences found across studies may reflect the duration of the study or outcomes studied.

Another not fully understood host-microbe relationship is that of the well-known bacteria *Akkermansia muciniphila* [190], which in absence of glycan DF [47], degrades the host’s mucosa-associated mucus layer, thus regulating mucus layer thickness. While *Akkermansia* is a mucin-degrader and a producer of propionate, acetate and ethanol [191], it is overrepresented in faecal samples from healthy individuals when compared to disease cohorts [190]. *Akkermansia* up-regulates the *Muc2* gene in human enterocytes, increasing the amount of mucus produced [47], which may lead to a thicker mucus layer in the presence of glycan DF, thus assuring optimal barrier properties. By modulating the fucosylation status of mucus [192], *A. muciniphila* further regulates how other mucus-degraders such as *B. thethaitaomicron,* digest the protective mucus layer when dietary glycans are unavailable. Abundance of *Akkermansia* seems to decrease with age [193]. Interestingly, *Akkermansia* was similarly abundant in young adults when compared to elderly free from disease or centenarians in Italy [114]. Butyrate-producing bacteria were positively associated with age, with *Eubacteriium limosum* overexpressed in centenarians when compared to the other arms of this cohort [114]. Depletion of *Akkermansia muciniphila* has also been associated with mass translocation of endotoxin-activated CCR2+ monocytes, leading to pancreatic injury and type 1 diabetes (T1D) [194]. The examples of *Akkermansia* and *E. limosum* may reflect an intricate symbiont homeostasis driven by diet.

Thus, stratification of human populations based on their relative microbiota abundance species is challenging. In this regard, the concepts of eubiosis and dysbiosis have been introduced. Eubiosis refers to a still undefined, but balanced and adequate GM population. Dysbiosis corresponds to a dysfunctional GM, which may start developing as early as during the neonatal period [195]. Whether dysbiosis can be a reaction to disease or instead, drive disease, is yet to be determined [180]. In a meta-analysis by Duvallet et al., it was found that dysbiosis can be further categorized, i.e., a dysbiotic state relating to an increase of pathogenic bacteria, vs. a dysbiotic state in which health-associated bacteria are reduced or missing [188]. One may refer to these states as inflammatory dysbiosis and hypotrophic dysbiosis, respectively. Microbial signatures have previously been shown to be disease-unspecific, i.e., there does not seem to be a direct association between specific bacteria and concrete pathologies [196]. In a dysbiotic state associated with disease (e.g., T2D, IBD), *A. muciniphila* is typically reduced in number [190]. This may result from a positive feedback loop, where lack of mucus (firstly due to possible lack of DF as alternative energy source, secondly due to increased microbial competition and/or decreased mucus production [47]) impedes *A. muciniphila* to reproduce, which in turn diminishes the *Akkermansia*-derived mucugenic and tolerogenic signals. This may further downregulate mucus production, allowing for pathobiont invasion of gut laminae and the pro-inflammatory *milieu* predisposing to disease development. Thus, during a prolonged time-trajectory, hypotrophic dysbiosis may be a gateway for inflammatory states [197]. For example, the infection by the parasite *Giardia lamblia*, endemic in several regions of the world, alters the GM due to its metabolites. Hypotrophic dysbiosis, and not directly through eliciting an inflammatory response, may be the cause of diarrheal states associated with giardiasis [198].

**Table 2 nutrients-14-05361-t002:** Identified microbiome signatures (DF fermenters) in health and disease (i.e., eubiosis and dysbiosis). In eubiosis, mean relative abundance (~98% bacteria retrieved) of phyla: 60% (58–88%) Firmicutes (F); 22% (8.5–28%) Bacteroidetes (B); 5% (2.5–7%) Actinobacteria (A); 5% (0.1–8%) Proteobacteria (P).

SCFA(s)	Bacterial Genera (Phylum)	Representative Bacterial Species	Observed Effects	References
Butyrate	Clostridiales cluster I-II (F)	*Clostridium histolyticum*	Identified as a potential tumour regression therapy (via collagenase production) as well as being associated with gas gangrene in diverticular disease and trauma (via exotoxin)	[199,200]
	Clostridiales XIV, Ruminoccacea (F)	*R. bromii*	Taxon driver of enterotype 3; Believed to be the main resistant starch fermenter into butyrate, was significantly increased following RS diet in men with obesity	[168,201,202]
	Clostridiales XIV (F)	*Clostridium symbiosum*	A SCFA producer, was shown to improve post stroke disability in aged mice	[203]
	Clostridiales IV, Lachnospiraceae (F)	*Roseburia intestinalis* *Butyrivibrio fibrisolvens*	Can rescue intestinal epithelium autophagy and mitochondrial respiration insufficiency, are associated with reduced colorectal cancer; *Lachnospiraceae* phylotypes increased on an NSP diet with strong cross-feeding interactions	[73,79,202,204]
	Clostridiales IV (F)	*F. prausnitzii*	Produce butyrate in 1 step reaction; Influences *Muc2* and goblet cell differentiation; depleted in IBD and Crohn’s disease	[52,205,206]
	Eubacteriae (F)	*E. rectale*, *E. hallii*, *E. ventriosum*	Together with *F. prausnitzii*, are the major butyrate producers; growth is promoted by low colonic pH, which also inhibits pH-sensitive pathogenic bacteria	[207,208]
	Propionibacteria (F)	*P. acidipropionici*	Propionate producer, induces colorectal cancer apoptosis through mitochondrial adenine nucleotide translocator (ANT)	[57,63,209,210]
	Bacteroides (B)	*B. thetaiotaumicron*	Driver of enterotype 1; is a mucus-forager with lack of DF *B. thetaiotaumicron* regenerates NAD+; reduced S-BCAA and alleviated diet-induced weight-gain and obesity in mice. Influences *Muc2* and goblet cell differentiation. Produces butyrate via the succinate pathway	[168,196]
Propionate	Negativicutes (F)	*N. succinicivorans*	Produce propionate via succinate pathway	[211,212]
	Veillonellaceae (F)	*V. parvula*	Produce propionate via acrylate pathway (lactate) and/or acetate. Have been associated with osteomyelitis, hypertension and endocarditis	[211,213,214]
	Lachnospiraceae (F)	*Blautia hydrogenotrophica*	Produce propionate via acrylate pathway (lactate) and propanodiol pathways (deoxi-sugars)	[211,213,215,216]
	Christensenellaceae (F)	*C. minuta*	Regarded as the most heritable taxon, forming the hub of a co-occurrence network composed of other heritable taxa; is enriched in lean subjects; in mice, reduced adiposity gain in GF model	[217,218,219,220,221]
	Bacteroides (B)	*B. fragilis* *B. ovatus*	Ferment xyloglucans, C3 directly inhibited Salmonella overgrowth by pH modulation in vitro. Bacteroidetes relative abundance has been linked to faecal propionate concentration. Decreased in ASD; in contrast, C3 administration led to ASD behaviour in rodent models via altered mitochondrial metabolism	[218,219]
Acetate	Prevotella (B)	*P. intestinalis*	Driver of Enterotype 2; significant high prevalence of Prevotella in healthy African Americans 50–65 y, while decreased in Western populations. *P. intestinalis* administration in mice led to reductions of overall SCFA production and increased mucosal inflammation which abated with IL-18 supplementation	[70,168,222,223,224,225,226,227]
	Methanobrevibacter (F)	*Methanobrevibacter smithii*	Found to be highly inherited, methanogens are inconclusively associated with increased BMI and reduced transit time in humans, as well as with leanness in mice. Metabolizers of formate, which can result in decreased blood pressure. Co-culture with *R. intestinalis* and *B. hydrogenotrophica* decreased H2 and produced CH4 and acetate, reducing pH	[196,202,204,217,218,219,220,221,228,229]
	Bifidobacterium (A)	*B. adolescentis*	FOS, GOS fermenter. High inheritability LF diet with prebiotic supp. increased *Bifidobacteria* abundance, which ameliorates the allergic phenotype and inhibited the growth of enteropathogenic bacteria. *Bifidobacteria* seems to be reduced in obese-derived faecal cultures as well as in ASD; Significantly decreased with a weight-loss diet given to men with obesity	[70,196,202,220,224,225,226,227,228]
	Lactobacillus (B)	*L. johnsonii*	*Lactobacillus* is a lactate producer commonly found in the upper gastrointestinal tract. FOS, GOS fermenter. May protect against diet-induced obesity and reduce asthma incidence in children. However, is increased in ASD. Probiotic supplementation impact revealed to be dependent on basal microbiota between individuals with obesity and normal weight	[70,168,220,222,223,224,225,226,228]

### 3.2. Genotypes—Interactions with Gut Microbiota

The interplay between the host genome and the microbiome is complex and dynamic [230]. While the enterotype may be subject to change over a lifetime [5,115], being affected not just by diet [231], but also by lifestyle factors such as smoking status [232,233], exercise or geographical location [176], the host genome is considerably more stable [3,234,235], although epigenetic modifications may occur in relation to environmental exposure.

Nutrigenetics focuses on how individual genetic profiles, such as copy number variation (CNVs) and single nucleotide polymorphisms (SNPs) may influence fates of different food items through, e.g., absorption, distribution, metabolism or excretion (ADME) patterns (Table 3). In recent years, CNVs and SNPs in coding and non-coding regions of the genome were identified as drivers of phenotypical differences among individuals. Polygenic risk scores have been associated with phenotypes across various human pathologies [236,237]. When describing homeostasis on the holobiont level (this is, when taking the host genome and microbiome together) [230,238], polygenic risk is thus relevant [239].

Twin studies have indicated that certain SNPs may be affecting microbiota colonization since implantation in the first days of life [96,240], acting as a matrix where optimal homeostasis and adaptation to environment will be grounded. A number of SNPs that may interact with SCFA, such as G-protein coupled cellular receptors (*GPCR41*, *GPCR43*, *GPCR109A*) [99], transporter encoding genes such as *MCT*, *SMCT* (monocarbohydrate transporters), effector genes such as *MUC2* (for mucus layer production in colon) or regulatory genes such as *NRF2* (regulating the expression of proteins involved in the bodies’ antioxidant defence mechanism such as superoxide dismutase (SOD)) may, independently or in polygenic aggregation, predispose to different health outcomes in human populations.

In studying elderly populations, genes responsible for inflammatory response such as *IL-6*, *IL-10* and the *IL-1* cluster, genes involved in the insulin/*IGF1* pathway and genes involved in oxidative stress management (*PON1*) were correlated with extreme old age [112]. Indeed, while bacterial colonization in early life is essential for the correct development of MALT germinal centres, NK cell maturation and Treg differentiation, establishing a balance between pro- and anti-inflammatory T cell subpopulations in the mucosa as reviewed elsewhere [5], the host’s genetic background was shown to modulate the extent of bacterial effects. Protein programmed cell death (*PD1*) knockout mice models have developed certain modified forms of immunoglobulin A (IgA), which led to altered microbiota profiles, specifically by reducing the numbers of bacteria from the genera *Bifidobacterium* and *Bacteroides* and increasing the bacteria belonging to the family *Enterobacteriaceae* [241]. Adequate secretion of IgA is essential for the colonization of certain “good” bacteria, while at the same time targeting “bad” bacteria, further deepening our understanding of the interdependency of host genome and microbiome. Commensal bacteria require IgA coating for colonization, while the same coating leads to immune responses towards pathogenic bacteria [242,243]. Interestingly, it was found that microbial acetate in the gut regulated IgA reactivity to commensal bacteria, thus selecting microbiota species and its colonization of the colon [244].

Furthermore, antimicrobial peptides (AMPs), such as α-defensins and β-defensin 1, were produced by intraepithelial cells (IEC) in the gut after stimulation by IL-22 and IL-17 as a way of quickly inactivating breaching microorganisms. AMPs not only helped to sustain host–microorganism segregation, but affected microbial composition [245]. Mice deficient in *MYD88* (an important member of the Toll/IL-1 receptor family [246]), *NOD2* [247] (a gene associated with intestinal homeostasis and IBD [248]), or mice transgenic for α-defensin 5 [249], exhibited an altered microbiota composition. In regard to the genotype of Alzheimer’s disease, strongly linked to the APOε4 allele, associations between higher levels of *Erysipelotrichaceae*, a family including pro-inflammatory bacteria was found, while the protective APOε2 allele was positively correlated with family *Ruminococcaceae* (SCFA producers). It was further shown that SCFAs are able to inhibit amyloid β (Aβ) aggregation—the histopathologic hallmark of Alzheimer’s disease—in vitro. Mutations in *MEFV* (leading to familial Mediterranean fever) [250] have also demonstrated the ability of the host to modulate microbial composition. This suggests that the host genome and microbiome structure may be functionally linked, striving for homeostasis. The host genotype would thus constitute another level of inter-individual variability towards DF effects, both directly (i.e., metabolism of SCFA) and indirectly (i.e., by modulating GM) [217].

**Table 3 nutrients-14-05361-t003:** Identified possible host genetic variability related to SCFA-ADME steps and effects in blood and tissues.

Metabolic Step (Tissue)	Gene (Protein)	SNP/CNV *	Observed Statistically Significant Association from GWAS	References
Digestion enzyme (gut lumen)	AMY1/2	CNV rs370981115	Impacts oral and gut microbiome due to bioavailability of starches; altered blood protein measurements	[251,252]
	LCT	rs4988235, rs1446585, rs2322659, rs35837297	Lactase persistence allows for dairy product consumption in adult life and increased expression of Bifidobacterium in the gut; altered lung function and leukocyte counts	[253,254,255]
Barrier function (colon)	MUC2	rs4077759, rs10794281, rs35225972	Modulated by butyrate. Variations associated with decrease gastric cancer progression, enhanced gastric lesion regression, asthma	[61,256,257,258,259]
	FUT2	rs516246, rs601338, rs679574	Mucus fucosylation status. Predisposition to Crohn’s disease and dysbiosis; altered blood protein measurements	[252,258,260]
Antimicrobial peptides (gut)	DEFA5	CNV rs2272719	α-defensins modulate microbial populations; copy number gain identified as pathogenic; altered white blood cell counts; susceptibility to paediatric leukaemia	[249,261]
	MMP7	rs11568818	Involved in antimicrobial processes; prostate cancer	
SCFA receptor	MCT1 (SLC16A1)	rs147836155 rs4839270 rs773430	SCFA uptake; variations have been associated to exercise-induced hyperinsulinemia (EIHI); microglial activation, refractive errors of the eye, blood pressure disorders	[262,263,264,265]
	MCT2 (SLC16A7)	rs79297227	SCFA uptake (hepatocytes); BMI trajectories, development of non-small cell lung carcinoma	[266]
	MCT3 (SLC16A8)	rs1004763	Cerebral white matter microstructure; cognitive function	[267]
	MCT4 (SLC16A3)	rs4239020	Adipose tissue distribution, BMI	[268]
	MCT11 (SLC16A11)	rs13342232	Associated with the risk of paediatric-onset T2D in Mexican families	[269]
	MCT9 (SLC16A9)	rs7094971	Carnitine transporter, associated with reversible ASD and mitochondrial abnormalities	[221,270]
	SMCT1 (SLC5A8)	rs7296340 rs141751904	SCFA uptake by colonocytes; in absence of microbiota, marked down-regulation of SLC5A8, which acts as a tumour suppressor protein in the presence of butyrate; variation decreases BMI-adjusted waist-hip ratio; decreased IL-2 levels	[271,272,273]
	SMCT2 (SLC5A12)	rs10835056	SCFA uptake; decreased MIP-1α levels	[273]
Metabolism	GPCR109A (HCAR2)	rs56959712	Butyrate receptor in enterocytes and MALT, regulating dendritic cell and Treg diff, also present in microglia. Ligand niacin is used to treat dyslipidaemia; variant associate with blood lipid measurements	[274,275]
	GPCR43 (FFFAR2)	rs34536858	Acetate and propionate receptor, leading to NLRP3 assembly. Regulation of Treg population in colon, ROS production and neutrophil chemotaxis. KO models showed increased arthritis, colitis and allergic disease; regulates adipogenesis and GLP-1 release; associated white and blood cell variance	[72,276]
	GPCR41 (FFAR3)	rs10407548	Regulation of SCFA-dependent energy homeostasis. Activation by propionate, butyrate and valerate results in inhibition of NF-κB activation; induce chemokine and cytokine expression; associated with gastrointestinal motility and stool frequency	[277,278]
	GPCR42	CNV	Recently reclassified as functioning gene; Propionate affinity; polymorphisms associated with strong pharmacokinetic variation	[279]
Metabolism (systemic)	LEP	CNV, rs7799039, rs17151919	40–70% estimated heritability for BMI; SNPs associated with CVD and MetS, increased HbA1c, insulin and increased fat mass, among other clinical phenomes. KO mice had higher susceptibility to dysbiosis	[280,281]
	LEPR	CNV, rs1137101, rs9436747	Same as above, variations associate with blood lipids, proteins, cytokines and cell counts	[280,282,283,284]
	PLD1	rs4894707	Associated with obesity, insulin sensitivity and abundance levels of *Akkermansia muciniphila*	[285]

* selected examples of SNPs, *p* < 5 × 10^−6^ as reported by the GWAS catalogue, not extensive. ASD, autism spectrum disorders; BMI, body mass index; CNVs, copy number variations; GWAS, genome-wide association studies; MIP-1α, macrophage inflammatory protein-1 alpha; SNPs, single nucleotide polymorphisms; LPS, lipopolysaccharide; GLP-1, glucagon like peptide 1; GLP-2, intestinotrophic proglucagon-derived peptide; CVD, cardiovascular disease; MetS, metabolic syndrome; KO, knock-out mice model.

### 3.3. Phenotypes, Epigenetic Aspects of SCFA

Complementary to nutrigenetics, nutrigenomics refers to how nutrients influence gene expression. In this regard, SCFA may directly influence genetic expression via histone deacetylase modulation [54,57,286,287]. Accumulated epigenetic variations such as histone (de-)acetylation, may translate into an individual’s phenotype over time. The phenotype refers to the observable, apparent properties resulting from the interplay between genetics, lifestyle and environment. As a result, disease and health status tend to be categorized according to interpretable anthropometric, clinical or laboratorial parameters, e.g., age, body mass index (BMI) or blood counts, respectively. Such outcomes might have different associations with the health status, according to the study conducted: epidemiological versus clinical and mechanistic studies. The variations observed in certain epidemiological studies may be due to genetic predisposition and phenotypical flexibility of the individuals, i.e., the capacity to maintain homeostasis and how to deal with environmental stressors [288,289,290]. Such flexibility may, on one hand, be epigenetically determined, via DNA methylation, histone acetylation or imprinting, which may be determined as early as during prenatal development [291], as seen in cases such as the Dutch Winter Hunger [292]. On the other hand, such flexibility may be related to genetic factors (i.e., nutrigenetics), as we will emphasize for the specific case of SCFA utilization and regulation of immunometabolism.

It must be noted, however, that such associations at the level of the phenotype (such as the relation between BMI and metabolic abnormalities, or the levels of glycated haemoglobin and diabetes progression), are possibly driven by genome, microbiome and diet interactions, which entail environmental, neurophysiological and hormonal factors. As some authors notice, a genotype presenting with variations leading to an increased level of inflammatory function (which may be relevant to defend against infections), may have deleterious effects when a low-grade chronic inflammation is not desired, such as happening with NCDs [17] and ageing [112], as shown in the association of IL-6 levels and T2D incidence [293]. Furthermore, even genes not related to immune function may have a relevant impact in attaining holobiont homeostasis, through interactions still not fully understood, which may lead to NCD and ageing progression, such as proposed for *FUT-2* [260] and *AMY1* [251].

Centenarians are individuals belonging to a group of people who reach ages over 100 years, without significant present chronic disease. Although research on those groups is in its early steps, some studies suggest that centenarians present with distinct GM signatures, when compared with young adults (20–40 years old) and elderly in general (60–80 years old) [110,114,115], although geographical differences were noticed across studies, regarding specific bacterial species’ abundances. Although other factors for variability exist, it has been argued that these results may reflect specific gene polymorphisms. The *FUT2* gene, encoding for fucosyltransferase 2, a protein present at the Golgi membrane that is associated with regulating the composition and function of secreted glycans in mucosal tissues of the gut and other tissues, being a predisposing factor for Crohn’s disease [260] is an example. Enterotyping further revealed that the (mucus) secretor phenotype was more likely to cluster in ET-3, or ET-F (associated with *Firmicutes*, *Akkermansia* and *Ruminococcus* spp.) [294]. This goes with the evidence that genes not directly involved in immune function may be responsible for several health outcomes, including the development of NCDs and survival into old age [295]. For example, copy-number variations (CNVs) in the salivary α-amylase (*AMY1*) seems to correlate with oral and GM composition [251], possibly via its role in carbohydrate digestion. In mice, the impact of the genotype in microbial colonization is well recognized [96]. On the other hand, the time of meals (i.e., chrononutrition) may also lead to significant changes in GM composition, as well as immune and metabolic conditions, including T2D [296,297], CVD [298] and psychological well-being [299]. While assessing the effect of DF intake timing on postprandial and 24 h glucose levels, stronger reductions in both *Ruminococcus* and 24 h glucose levels were found with morning DF ingestion, whereas the reduction of these phenotypes in the evening was less pronounced [300]. Another study found that diurnal oscillations of oral microbiota composition were linked to salivary cytokine levels, particularly IL-1β and *Prevotella*, and IL-6 with *Prevotella, Neisseria* and *Porphyromonas* [301].

Recent studies have emphasized the interplay of diet and GM in persons with genetic predisposition regarding neurodegeneration [11], cancer [35], menopause symptoms [302] and non-alcoholic fatty liver disease (NAFLD) [303], all of which are characterized by chronic inflammation, locally or systemically [24], although further research is needed [304] (Figure 1).

As mentioned above, Goodrich et al. [3,217] also found microbiota heritability in humans through studying homo- and dizygotic twins. As the microbiome is associated with health status and fitness (Table 4), the host may benefit from interactions between their genomic makeup and microbiome composition, which are modulated by dietary patterns among other environmental exposures [3]. A degree of inter-individual variability exists, in what concerns SCFA production. In a longitudinal study, while acetate was generally the most abundant SCFA in faeces of all individuals, one individual presented with a 10-fold decrease in propionate and butyrate when compared with other participants. One individual presented with high caproate concentrations across the observation. Traces of valerate were consistently detected in all individuals [305]. In this study, microbiota profiles remained stable, and the only unpredictable variable was ammonium concentration in stool.

In the following chapters, we will attempt to elucidate further the host-microbe driven interactions associated with SCFA.

**Table 4 nutrients-14-05361-t004:** Identified dysbiosis signatures in disease.

Condition(s)	Increased Bacteria	Decreased Bacteria	Opportunistic spp. or Additional Findings	References
Obesity	↑Firmicutes:Bacteroidetes ratio, Blautia, Dorea, Proteobacteria, Tenericutes	Akkermansia, *F. praustnizii*, *B. thetaiotaumicron*	Ratio seems to be higher in women with ↑BMI Diversity and richness is crucial for responding or not to dietary intervention aiming at improving metabolic parameters (insulin sensitivity, lipid and inflammation markers); increased propionate production compared to normal weight microbiota	[196,202,306,307,308]
Metabolic Syndrome	↑Firmicutes:Bacteroidetes ratio, Blautia, Dorea, Methanobacteriaceae	Oscillospira, Rikenellaceae, Bifidobacterium, Christensenellaceae, Akkermansia, Lactobacillus	BCFA are associated with obesity, insulin resistance and development of T2D; *Bacteroides* spp. may improve the efficiency of BCFA degradation Ass. With ↑faecal SCFA, plasma BCFA, plasma TMAO, plasma total bile acids and plasma LPS. MetS and NAFLD seem to occur via intestinal FXR	[196,309,310]
Gestational diabetes	Collinsella, Rothia, Desulfovibrio, Faecalibacterium, Anaerotruncus	Clostridium, Veillonella, Akkermansia, Christensenella	Similar findings with obesity enterotype, may remain postpartum *P. copri* and *B. vulgatus* identified as the main species leading the biosynthesis of BCFAs and insulin resistance; prebiotic supp. increased *Bifidobacteria* and led to reduction of faecal SCFA and serum fasting glucose and insulin	[196,306,311]
T2D	↑Firmicutes:Bacteroidetes, Dorea, Escherichia, Clostridiales, Lactobacillus	Overall diversity reduced; *R. intestinalis*, Akkermansia, Streptococcus, Bifidobacteria, *F. prausnitzii*	Similar findings with obesity and MetS enterotypes, although some studies find ↑Bacteroidetes:Firmicutes ratio. Opportunistic infections with *B. caccae, C. hathewayi, C. ramosum, C. symbiosum, E. lenta* and *E. coli*. Butyrate is beneficial for pancreatic B-cell function, whereas propionate has shown to be detrimental. Metformin therapy increases *A. muciniphila*	[196,308,312,313]
T1D	↑Bacteroidetes:Firmicutes, Synergistetes	Clostridium, Prevotella, Bifidobacterium Lachnospiraceae, Veillonellaceae	Opportunistic overgrowth of *Ruminococcus gnavus* and *Streptococcus infantarius*. T1D may be related to delivery method, feeding method and antibiotic use in infancy	[314]
NAFLD	Lactobacillus, Dorea, Streptococcus, Lachnospiraceae	Ruminococcaceae, Prevotella, Flavobacterium, *B. vulgatus*	Increased intestinal permeability associated with the degree of steatosis, affects up to 70% of patients with T2D and 90% of obese, possibly due to intestinal inflammation and permeability dysfunction, bile acid metabolism (FXR), anaerobic fermentation, and LPS activation of TLR4 leading to insulin resistance	[196,310,315,316]
Non-alcoholic Steato hepatitis (NASH)	Bacteroidetes, Prevotella, Escherichia	Firmicutes	Prevotella seems to be reduced in advanced stages of NAFLD, i.e., NASH; the levels of serum LPS and TNF-α correlated with disease severity. Synbiotic supp. of *B. longum* and FOS reduced disease severity of NAFLD and NASH progression	[196]
Alcoholic Steatohepatitis	*E. faecalis, E. coli*, Proteobacteria	Bacteroidaceae, Ruminococcaceae, Firmicutes	Only 40% of patients had dysbiosis. *E. faecalis* correlated with mortality rates in alcohol-induced steatohepatitis; supp. with *B. subtilis* and *E. faecium* improved symptoms and microbiome	[317]
IBD	Proteobacteria	Firmicutes, esp *F. prausnitzii*; Bacteroides; Clostridium; Peptostreptococcus; Bifidobacterium	Increase in fungal *Candida albicans, Aspergillus clavatus*, and *Cryptococcus neoformans*, decreased *Saccharomyces cerevisiae*. IBD can arise from genetic susceptibility or from disruption of commensal bacteria such as SCFA-producing bacteria, reduction in tryptophan metabolism (promoting mucus barrier function and reduces inflammatory responses), Proteobacteria may represent 20% of overall diversity	[35,258]
Colorectal cancer	*S. bovis*, *H. Pylori*, *E. faecalis*, *E. coli*, *B. fragilis*, *F. fucleatum*, *C. septicum*, Fusobacteria, Proteobacteria, Akkermansia	Bifidobacteria, Lactobacilli, Bacteroidetes, Firmicutes, *F. prausnitzii*, Prevotella, Porphyromonas	*S. bovis* is increased in neoplastic milieu and may forage tumour metabolites, inducing inflammation. Some bacterial strains may propel CRC development, while others are only found in late stages of CRC, arising as opportunistic pathogens, which may deplete symbionts by substrate competition and lead to tumour survival by immune evasion mechanisms.	[16,50,52,57,210,318,319]
Psoriatic arthritis	N.A.	Coprococcus, Akkermansia, Ruminococcus, Pseudobutyrivibrio.	Overall reduced microbial diversity, similar to IBD and other autoimmune phenotypes such as skin psoriasis; however, Akkermansia and Ruminococcus were uniquely decreased in psoriatic arthritis. Rheumatoid arthritis presents with increased *P. copri*	[320]
Atopy, inc. food allergy, atopic dermatitis and asthma	↑Firmicutes:Bacteroidetes, *C. difficile*, Enterobactericeae, *E. coli*	Bifidobacteria, Lactobacilli, Clostridia, Bacteroides, Actinobacteria, Proteobacteria	Supp. *L. rhamnosus* GG and *L. fermentum* to mothers in the prenatal and early postnatal periods or to young children may be effective in reducing symptoms, treatment and prevention of early atopic disease in offspring	[321,322,323]
Autism Spectrum Disorders	Clostridium, Bacteroidetes, Lactobacillus, Caloramator, Sarcina, Propionibacteria, Desulfovibrio	Bifidobacterium, Prevotella, Firmicutes, Akkermansia	Increased production of propionate due to dysbiosis may be a cause of reversible ASD, also leading to GI symptoms in a majority of cases, which ameliorated by supp. strains of *Bifidobacteria* and *Lactobacilli*. Children with ASD show increased levels of opportunistic *Candida albicans*	[324]
Cardiovascular Disease (CVD) inc. Atherosclerosis and Hypertension	↑Firmicutes:Bacteroidetes, Enterobacteriaceae, Clostridia (*C. histolyticum, C. perfringens, E. timonensis*), Atopobium, Prevotella	microbial richness, diversity and evenness significantly decreased, Odoribacter, Bacteroides	S-TMAO (microbial-derived choline metabolite) levels were dose-dependent associated with CVD outcomes and other indicators such as serum cholesterol, glycaemic indices (HbA1c, fasting plasma glucose), inflammation biomarkers (IL-6, CRP), overall cardiovascular risk, and metabolic syndrome.	[90,196,325,326]
			Odoribacter is a butyrate-producer negatively correlated with systolic blood pressure, like other SCFA producers, although SCFAs increase vascular tone	[308,325,327,328,329,330]
Parkinson Disease	Bifidobacterium, Pasteurella, Enterococcus, Lactobacillus, Verrucomicrobia (*A. muciniphila*), Bilophila, Christensenella, Dorea, Barnesiellaceae, Tissirellaceae, Ralstonia, Pasteurellaceae. Escherichia, Bacteroidetes	Firmicutes, Brautella, Prevotella, Faecococcus Lachnospiraceae, Paraprevotella, Faecalibacterium, Roseburia, Blautia, *C. coccoides, B. fragilis*	Paraprevotella mainly decreased in females; Bilophila abundance associated with disease severity; Blautia associated with disease onset/duration; neurotransmitters such as serotonin, dopamine and GABA are produced by microbiota; *E. coli* producing amyloid protein Curli cross-seeds with α-synuclein and stimulates protein aggregation in gut (present in 65–85% of cases), with gut-to-brain transport demonstrated. Microbial sulphur metabolism is profoundly changed in PD, mainly associated with *A. muciniphila* and *B. wadsworthia*	[11,12,31,331,332,333,334,335,336,337,338,339]
Alzheimer’s Disease (AD)	↓Firmicutes:Bacteroidetes, *E. coli*, Shigella, Helicobacter, Odoribacter	Bifidobacteria, Lactobacillus, Firmicutes, Actinobacteria, Verrucomicrobia, Roseburia, Eubacterium, *F. prausnitzii*	Similar dysbiosis in MCI as in AD; amyloid protein Curli produced by *E. coli* and *S. typhimurium* enhances colonization and biofilm development; *E. rectale* and Shigella taxon in the faecal samples of patients with advanced AD correlated well to the amyloidosis and level of proinflammatory cytokines in the brain. TMAO induced synaptic impairment in AD model with deposition of Aβ plaques and neurofibrillary tangles. Aβ plaques found in gut vessels prior to disease onset, accompanied with systemic inflammation	[11,12,14,90,334,339,340,341,342]

Abbreviations: Aβ, amyloid-β; AD, Alzheimer’s disease; BMI, body mass index; BCFA, branched-chain fatty acids; CNS, central nervous system; CVD, cardiovascular disease; FXR, Farnesoid X receptor; GABA, γ-aminobutyric acid; GI, gastrointestinal; HDL, high density lipoprotein; IBD, inflammatory bowel disease; MCI, mild cognitive impairment; MetS, metabolic syndrome; LPS, lipopolysaccharide; SCFA, short chain fatty acids; Supp., supplementation; T2D, type 2 diabetes mellitus; T1D, type 1 diabetes mellitus; TMAO, trimethylamine N-oxide.

## 4. The Holobiont and Short Chain Fatty Acids

### 4.1. Host-Microbe Interface

The human gut is composed of several tissues with specific characteristics. Whereas the relatively short passage time (up to 2–3 h) and low pH of the stomach (1.5–2 in fasting state, up to 5 with meal [343]) are associated with low numbers of bacteria (<10^2^/mL) [4], the slower passage and more stable pH progressively attained in the small intestine may allow for an increase of >10^8^/mL microbes at the ileal-cecal valve. In the colon, the bacterial community increases gradually from proximal to distal, with viable cell counts in faecal samples, reaching 10^11^ to 10^12^ cells/g, the majority being obligate anaerobes [4,63].

Of interest is the change in luminal pH, which modulates significantly which bacterial species can colonize different gut territories. The increase in luminal pH occurs mostly due to neutralization of gastric acids and pancreatic secretions in the small intestine. Food intake is the main determinant for availability of DF, and its composition as a substrate will influence which fermentation products will be formed, depending on redox capacity [78,344]. An increased number of DF fermenters proximal in the colon will reduce luminal pH (down to pH 5.5–7.5) [345], increasing GM diversity in the colon, as individual species of microbiota use each other’s complex carbohydrate breakdown products (substrate cross-feeding) [216,346], or even end-products (metabolic cross-feeding) in mutualistic interactions promoted by anoxic conditions common in the colon [347]. Den Besten and colleagues [348] showed that bacterial cross feeding occurs mainly with conversion of acetate to butyrate, to a lower extent from butyrate to propionate, and virtually no metabolic flux exists between propionate and acetate. Of note, *F. prausnitzii* is able to derive butyrate from acetate produced by *B. thetaiotaumicron.* This could have a significant impact on the intestinal barrier [205], as butyrate increases mucin production, resulting in increased *MUC3*, *MUC4* and *MUC12* gene expression, potentially through mitogen-activated protein kinase (MAPK) signalling pathways [349], as well as up-regulating the assembly of tight junctions through activation of AMP-activated protein kinase (AMPK) [350]. Both processes would enhance barrier functionality. While acetate produced by *Bifidobacteria* may play a role in inhibiting enteropathogenic microbial growth such as *E. coli* [224], another colitis mice model revealed that inoculation with *Lactobacillus rhamnosus* L34 profited both local gut inflammation (reduced leaky gut and faecal dysbiosis), as well as systemic inflammation, possibly due to reduced translocation of lipopolysaccharides (LPS) [351]. Further, *L. reuteri* GroEL protein administration also reduced markers of inflammation (TNFα, IL-1β, IFNγ) induced by LPS via TLR-4 both in vivo and in vitro [352].

SCFA derived from fermentation are at highest concentrations in the proximal colon, and their concentrations decrease towards the distal colon, as reviewed by Topping and Clifton [50]. Colonocytes progressively absorb acetate, propionate and butyrate, which enter the bloodstream. Their rate of absorption, as well as effects on crypt proliferation are dependent on luminal pH, as demonstrated by Ichikawa [353]. Specifically, butyrate is virtually absent in the portal circulation, and it has been found to be a main substrate for colonocyte energy requirements, affecting colonocyte proliferation and differentiation, as well as mucus production [63]. Blood butyrate may thus not be a good measure of butyrate production in the colon. Butyrate is at present regarded as a protective factor in the development of colorectal carcinoma, one of the leading causes of morbidity in developed countries [51,354]. Sometimes referred to as the “butyrate paradox” [355], it induces proliferation of healthy colonocytes, but terminates differentiation and triggers apoptosis in metaplastic cells via the Warburg effect [356]. SCFA concentrations change across the lumen, and tend to be less prevalent in sites of highest absorption, i.e., the colonic crypts [66]. Contrarily, GM-derived formate has recently been found to promote colorectal carcinoma progression [357].

### 4.2. Digestive Enzymes

Poole et al. [251] demonstrated that the oral and gut microbiotae of individuals with normal BMI and no chronic disease may be related to the copy-number of the *AMY1* gene, encoding salivary α-amylase. Copy-number variations (CNVs) in *AMY2*, encoding pancreatic α-amylase, were positively correlated with salivary amylase copy-numbers. Looking to attain homeostasis, individuals with a low copy-number of *AMY1* (which facilitates starch digestion) had microbiomes with enhanced capacity to digest carbohydrates, including increased members of *Lachnospiraceae*, as well as *Akkermansia* and *Bifidobacteria*, and presented increased faecal CAZyme activities of the glycoside hydrolase and polysaccharide lyase classes, in line with more complex carbohydrates reaching the distal gut [251]. In contrast, individuals with a high copy-number *AMY1* did display a higher abundance of DF fermenters (increased abundance of *Ruminococcus*, *Oscillospira* and *F. prausnitzii*), possibly to counterbalance for the absence of non-digested carbohydrates due to more optimal host digestion, as shown by increased SCFA concentrations in the stool. Using the enterotype stratification, both genotypes would fall into ET-3 (*Firmicutes*), although with substantial species differences. Why high copy-number *AMY1* individuals’ stool had increased concentrations of SCFA remains elusive, assuming that the rate of absorption across individuals is comparable. Possibly, bacterial cross feeding [67,216,346] is enhanced in low *AMY1* subjects, which could imply a more intense utilization of digestible starch as opposed to indigestible fibres as a source of microbial energy, which would in turn be related to lower overall rate of SCFA production. After faecal transplantation to germfree mice, mice receiving high *AMY1* individuals’ stool showed increased weight gain, although dietary intakes and gut inflammation parameters between mice were not significantly different. This may suggest that a high *AMY1* CNV predisposes the GM to more specialized digestion of RS, and possibly other types of fibre. It also suggests that an increased functional diversity between mutualistic bacteria (in subjects in which a higher proportion of complex carbohydrates reaches the colon, or low *AMY1*) was protective against weight gain, at least in mice [251]. This finding contrasts with human studies, which found an inverse association between *AMY1* numbers and overweight/obesity in elementary school aged children in the USA [358]. We can hypothesise that in this case, the microbiota of children with low *AMY1* may have already suffered adaptations regarding carbohydrate digestion capability.

Investigating the relationship between single nucleotide polymorphisms (SNP) and microbiota, Blekhman et al. [240], found that variants in the *LCT* gene (encoding lactase) were significantly correlated with *Bifidobacterium* in the gut. This correlation is interesting, as lactase persistence may permit individuals to continue consuming dairy products into adulthood, and certain products may contain *Bifidobacteria*. Indeed, in another study the *LCT* locus associations to GM composition seemed modulated by lactose intake, whereas others associations could be explained by secretor status as determined by the participant *FUT2* genotype [359]. This relationship makes the study of nutrigenetics and nutrigenomics all the more relevant, as it may help understand human dietary habits, as well as comprehend inter-individual variability of the GM. Furthermore, the human responses to medications containing lactose moieties, such as alprazolam, lorazepam, carvedilol or cetirizidine hydrochloride may be affected by such SNPs [240], which may result in different medication responses.

### 4.3. Genetic Diversity and Physical Barriers

#### 4.3.1. Mucin

Microbial communities can regulate the expression of the host’s physical barrier in the gut, particularly that of mucin production [205]. Mucin is the main glycoprotein component of the mucus layer that separates enterocytes and microbiota in the lumen; the mucin family is composed of 21 members. While mucin acts as a fundamental part of the mucosal barrier throughout the gut, different types are expressed in different gut tissues [360]. Recently, in order to understand the development of gastric cancer, the third leading cause of cancer-related deaths worldwide, polymorphisms in mucin genes were explored [360]. The gastric mucosa normally expresses Muc1, Muc5AC and Muc6. However, during gastric carcinogenesis, these were differently expressed, and Muc2 was concomitantly activated and secreted. It must be noted that *MUC2* is a major mucin gene in the intestine, particularly the colon, where the environment is profoundly different (pH, bacterial colonies) than that of the stomach. Furthermore, gastric carcinoma is associated with *H. pylori* infections; however, only 1–3% of infected persons develop gastric carcinoma, suggesting that SNPs in *MUC* genes may confer protection or risk for cancer. In effect, Muc2 is up-regulated in intestinal metaplasia. Marín et al. reported that three SNPs (rs10794293, rs3924453 and rs4077759) at the 3′ moiety in *MUC2* were associated with a decreased risk of lesion progression. Furthermore, four SNPs (rs10902073, rs10794281, rs2071174 and rs7944723) at the 5′ moiety of *MUC2* were significantly associated with regression of gastric lesions [257].

#### 4.3.2. Tight Junction Proteins (TJPs)

As a main source of energy for colonocytes, SCFA also help maintain the gut barrier integrity (as well as the blood-brain barrier) [361], by upregulating tight junction proteins (TJPs) such as claudin-5 and occludin [135]. Enteric bacteria with pathogenic potential can interrupt the impermeability of the gut, allowing for pathogen invasion of intestinal tissues and possible translocation into the host’s system. Such tissue invasion triggers an inflammatory cascade that has been associated with obesity and insulin resistance [362]. It has been recently shown that SCFA, particularly butyrate, promote recovery of tight junctions during gastrointestinal infections [363], possibly through mediation of different kinases (e.g., PKC [364], MAPK, PKA [365]), leading to phosphorylation of zonula occludens-1 (ZO-1) and inhibition of zonulin, increased expression of claudin-1 and occludin redistribution [366,367], reducing intestinal permeability [368]. Bacterial lipopolysaccharide (LPS) triggers a toll-like receptor 4 (TLR4) mediated pro-inflammatory cascade in mucosal immune cells, leading to the activation of signalling pathways, such as nuclear factor κ B (NF-κB) and MAPK, which promotes inflammation driven by cytokines such as tumour necrosis factor α (TNF-α) and IL-6. Inhibition of HDACs by butyrate results in reduction of LPS-induced activation of the NLRP3 inflammasome and autophagy and alleviates disruptions of ZO-1 and occludin, thus enhancing intestinal barrier function [369], as well as through repression of claudin-2 formation [187]. Butyrate may thus aid in counteracting negative effects of LPS induced pro-inflammatory cascades.

A permeable intestinal barrier has been associated with coeliac disease, inflammatory bowel disease, obesity and food allergies [370], apart from low-grade chronic inflammation and systemic disease development such as arthritis [9,371]. This mechanism was shown to be targetable in order to prevent the onset of arthritis, reduce disease progression and associated low-grade chronic inflammation, using butyrate or larazotide acetate [371]. Furthermore, it is recognized that the onset of Parkinson’s disease may occur in the gut, with α-synuclein aggregation upon LPS binding, eliciting a potent inflammatory response by specifically activating TLR4/NLRP3 inflammasome pathway. In addition, TLR2 has been shown to be important for the regulation of intestinal barrier integrity, being activated by different bacterial amyloid peptides. Through vagal axonal transport, these amyloid peptides are hypothesized to serve as a scaffold for the development of cerebral amyloid β aggregation (seen in Alzheimer’s disease) and α-synuclein aggregates (of Parkinson’s disease)[11]. TLR2 is also involved in neuroinflammatory process of clearance of amyloids such as α-synuclein and amyloid β [11,334].

#### 4.3.3. Immune Cell Populations in the Gut

Butyrate further impacts intestinal macrophages differentiating into M2 type macrophages (tolerant macrophages), which induce dampened responses to LPS stimulation and suppressed pro-inflammatory cytokine (IL-6 and IL-12) responses, via histone deacetylases (HDACs) [54].

Natural killer (NK) cell differentiation requires IL-23 produced by activated myeloid and epithelial cells, as well the presence of intestinal microbiota, as evidenced in germ-free vs. conventional mice studies [372]. NK cells produce IL-22, promoting a rather impermeable intestinal barrier [373] via signal transducer and activation of transcription 3 (STAT3). The GM also modulates the abundance of invariant NK T cells, a pro-inflammatory subset of T cells that secretes T helper 1 (T_H_1)- and T_H_2-type chemokines and cytokines, including interferon-γ, IL-2, IL-4, IL-13, IL-17A, IL-21 and TNFα [374]. The colon of germ-free mice is rich in invariant NK cells, further suggesting the immune-tolerant role of the microbiome through SCFA [57,59,286,375]. In human populations, evidence is mounting that antibiotic exposure in early ages [376] can predict risk of asthma development several years later. An opposite exposure, the so-called “farm effect” where many microorganisms may colonize the infant, are linked with reduced risks of asthma, atopy and possibly even autism spectrum disorder [95,377]. A study assessing the impact of prebiotic GOS in the elderly population found that GOS significantly increased the abundance of *Bifidobacteria*, at the expense of less beneficial taxa compared with the baseline and placebo arm. Phagocytosis, NK cell activity, and the production of IL-10 were significantly increased, whereas the production of pro-inflammatory cytokines (IL-6, IL-1β, and TNF-α) were reduced. GOS did not alter total cholesterol or HDL-cholesterol production, however [378].

#### 4.3.4. Transporter Genetics

SCFA are taken up by colonocytes, using proton and Na^+^-coupled monocarboxylate transporters (MCTs and SMCTs, respectively). Fourteen transporters belonging to the *SLC16* transporter family have been identified, of which four (*SLC16A1*, *SLC16A3*, *SLC16A7* and *SLC16A8*), encoding for MCT1, MCT4, MCT2 and MCT3 respectively, have shown to mediate proton-linked transport of monocarboxylates [379,380]. They allow for the uptake of carboxylated pharmaceuticals, as well as monocarboxylate transfer through tissues. The two members of SMCTs, *SLC5A8* and *SLC5A12*, are present in the gastrointestinal tract, kidney, thyroid, brain and retina [271].

MCTs 1–4 have distinct properties and tissue distribution, making them involved in a myriad of metabolic functions such as energy metabolism (specifically in the intestines, brain, skeletal muscle, heart and tumour cells), drug transport, thyroid hormone metabolism (*SLC16A2* or MCT8), and T-lymphocyte activation. This family of transporters has been studied in recent years, and recently reviewed [381]. Exercise-induced hyperinsulinemia, an autosomal dominant condition, has been attributed to a mutation in the promoter region of MCT1 in β-cells in the islets of Langerhans, leading to inappropriate insulin expression [262]. MCT2 appears to be an early indicator of prostate malignancies and MCT4 was associated with poor prognosis of prostate cancer, as reviewed elsewhere [382]. Nuclear localization of MCT1 in soft tissue sarcomas is instead associated with lower neoplastic scores and longer survival rates [383,384]. In retinal tissue, MCT3 and MCT4 seem to be drivers of correct cellular differentiation upon healing [385], and may thus be relevant in age-related retinal pathologies. Ongoing studies suggest that other members of SLC16A, such as MCT9, being associated with carnitine efflux [270], are a potential cause of reversible autism spectrum disorders [221]. Four missense SNPs, i.e., nucleotide polymorphisms leading to incorrect amino acid expression and one synonymous variant (Leu > Leu) on MCT11 were significantly associated with the risk of adult and paediatric T2D [381]. Thus, it appears that SCFA uptake and distribution is potentially intertwined with, and influenced by, a number of genetic variations that relate to disease conditions.

In the intestine, MCT1 present in the basolateral membrane of colonocytes allows for passive transmembrane transfer of SCFA into the bloodstream [271,386], following its active transport into the cell by the SMCT *SL5A8*, expressed in the apical membrane of epithelia. Like MCT1 [387], *SLC5A8* may act as a tumour suppressor as it mediates the uptake of butyrate, propionate and pyruvate. Butyrate is converted to acetyl-CoA in normal colonocytes, providing energy and up-regulating histone acetylases (HATs). Conversely, in metaplastic processes, cells turn to aerobic glycolysis due to the Warburg effect. In such conditions, butyrate and propionate accumulate, leading to reduced genetic transcription (through decreased HDAC activity) [356,388]. However, if high metabolism cancerous cells become deprived of glucose, oxidation of fatty acids is activated, converting butyrate or propionate into acetyl-CoA. This could result in the rescue of metaplastic cells, through upregulating HATs. This mechanism may potentially be involved in the metaplastic-to-anaplastic process seen in cancer [100].

Although both pathways result in hyperacetylation of DNA, different genes are affected and expressed. For example, in colon metaplasia, HDACs regulate intestinal macrophage activity [54] as well as inhibition of colonocyte proliferation, and induction of apoptosis. Furthermore, while butyrate was the strongest influencer of colonic HDAC expression, propionate and valerate caused growth arrest and differentiation in human colorectal carcinoma cells. Acetate and caproate did not cause histone hyperacetylation in this tissue [388]. Drugs such as salicylates, γ-hydroxybutyrate, valproate or non-steroidal anti-inflammatory drugs, such as ibuprofen, act as blockers of *SLC5A8* function, and may reduce SCFA uptake by colonocytes [379]. Long-term use of these drugs may alter colonic intracellular physiology. Furthermore, a number of naturally occurring inhibitors of MCTs have been described, such as stilbene disulphonates (including DIDS and DBDS), phloretin (a natural phenol) and bioflavonoids such as quercetin [389]. Given that these compounds occur together with DF and thus with SCFA, their effect may be negligible, or instead provide low-grade stress known as xenohormesis. Indeed, these compounds are generally known to have antioxidant, anti-inflammatory and therapeutic effects [390].

MCT1, in particular, is expressed in cells of the intestine, the colon and the blood-brain barrier, and may be relevant in delivering pharmaceuticals and SCFA across these membranes. MCTs 1–4 require the binding of a transmembrane glycoprotein (either embigin or basigin) for their activity [386,391], otherwise they will accumulate in the Golgi apparatus. MCTs 1–4 can mediate either cellular influx or efflux, depending on the prevailing substrate and pH gradients. MCT2 has the highest affinity for monocarboxylates, followed by MCT1 and MCT4 (MCT3 is less well characterized). MCTs further transport lactate, pyruvate and ketone bodies [391].

MCT1, MCT2 and MCT4 expression was significantly altered with fasting, and was tissue specific [392]. MCT1 transcription in skeletal muscle and T lymphocytes may be up-regulated following AMPK activation by SCFA. Other mechanisms may involve increased cytosolic concentrations of calcium, which stimulates calcineurin to dephosphorylate and activate NFAT (nuclear factor of activated T cells), resulting in up-regulation of target genes influencing the cell cycle, apoptosis and angiogenesis. Other mechanisms seem to regulate MCT1 expression in several tissues in response to obesity, diabetes and thyroid dysfunction, however more research is needed [389].

These results suggest that MCTs play a critical role in modulating adequate energy supply to different tissues of the organism, particularly dependent on the availability of oxygen, glucose or ketone bodies. For example, MCT2 is upregulated in neurons following food deprivation and recovery from ischemia. In contrast, hypoxia reduced MCT2 expression in adipose tissue. Furthermore, MCT2 expression is upregulated in the brain by both insulin and IGF-1, through a post-transcriptional mechanism involving stimulation of the phosphoinositide 3-kinase (PI3) pathway [389]. MCT4 was reduced in all murine tissues upon 48 h of fasting [392]. Concurrently, MCT4 expression was increased in all tissues in response to hypoxia via hypoxia-inducible factor 1α (HIF-1α), further supporting its role in glycolysis [393]. MCTs may show preferential binding to different SCFA [391]. As MCTs are also used for drug delivery, their expression in different health and nutritional states must be taken into account for optimal therapeutic results.

Concomitantly to MCTs being pleiotropic (i.e., different expression and different downstream pathways) in mammals, SCFA concentrations also change across tissues. How SCFA and MCTs interact in particular disease states is still not fully understood in human populations. Such mechanistic insights are relevant in maintaining health as well as for SCFA use as therapeutic agents.

### 4.4. SCFA Metabolism

As recently reviewed by Van der Hee and Wells [100], SCFA are estimated to contribute to 10% of human energy requirements, where butyrate is the main source of energy for colonocytes and propionate is partially converted to glucose in the liver [64,394]. It is estimated that the GM produces 500–600 mmol/d SCFA, whereof 60% are acetate, 20% propionate and 20% butyrate, amounting to about 37 mmol/kg body weight acetate, 13 mmol/kg body weight propionate and 12,4 mmol/kg body weight butyrate [100]. In the human descending colon, SCFA concentration may reach 69–91 mmol/kg luminal content, with acetate accounting for 60–75% of total faecal SCFA. *Methanobrevibacter smithii* is present in 70% of humans, and is considered the main methane producer in the GM. Methane production in humans (assessed by breath test) has been significantly associated with higher BMI scores in obesity, and was further associated with constipation and antidepressant use [229]. In the presence of methane, elongation of propionate (a whole-body energy regulator) can produce valerate [395], of which little is known regarding health maintenance. Similarly, little is known for caproate, also present in small amounts in the gut, which is generated from butyrate, acetate and lactate [396] under appropriate conditions. Despite their small concentrations found in several studies, over time these SCFA may alter the concentrations of acetate, butyrate and propionate reaching human cells, with a cumulative impact for health status.

#### 4.4.1. Colon

Colonic gut epithelia absorb more than 95% of SCFA produced by the GM. Butyrate oxidation accounts for more than 70% of colonocyte energy production [58,356,397], although colonocytes can also oxidise glucose and glutamine. Both butyrate and propionate appeared to increase cellular proliferation rates, while acetate did not [130,398,399,400]. This further highlights different SCFA utilization in particular tissues in the human body.

Butyrate may be essential for enterocyte differentiation, as previously reviewed [100]. Mature, but not progenitor enterocytes are strong butyrate metabolizers. It was a shown that butyrate leads to an arrest of proliferation and induction of differentiation of enterocytes, primarily by FoxO3, but also by hypoxia-inducible factor α (HIF-α). FoxO3 is associated with cellular homeostasis and longevity [401]. In contrast, activation of FoxP3 in naïve CD4^+^ T cells was associated with Treg differentiation and a tolerant profile [402,403]. This observation further suggests an important role of SCFA in the epigenetic modulation of several transcription factors, such as FoxO3 and FoxP3, which may relate to cancer development and tissue healing. Butyrate further inhibited DNA-damaged cell proliferation via p53 [404]. In intestinal crypts, a diffusion gradient allows for butyrate to be metabolized by apical mature enterocytes, also leading to upregulation of zonula occludens 1 and occludin, while down-regulating claudin 1 and 2. This would result in a net reduction on intestinal permeability. Apical mature enterocytes also produce TGB-β promoting Treg differentiation and a tolerogenic profile, with increased levels of circulating IL-10 [100]. Butyrate may thus directly and indirectly (through colonocyte paracellular signalling) improve intestinal permeability and immune function. Here, butyrate may further suppress TNF-α, IL-6, and myeloperoxidase activity by preventing NF-κB activation such as exemplified in Küpffer cells of the liver [405].

However, arterio-venous studies have demonstrated a relative indifferent usage of SCFA by colonic tissue [406]. Furthermore, acetate may be the strongest stimulant of intestinal blood flow, and appears to regulate the brain-pancreas axis regarding insulin-release regulation [6]. Regarding propionate, evidence is mounting regarding its role on phasic colonic motility [407]. On a more systemic level, it has been previously demonstrated that SCFA have dose-dependent effects in vitro and in silico, becoming inhibitory of smooth muscle cell proliferation at non-physiological high doses [408]. MCT1 and MCT2 SNPs may lead to different absorption rates from the gut in individuals, and may be predictive of colorectal cancer outcomes [409]. Ketone bodies produced from β-oxidation of SCFA serve as precursors for lipid synthesis in human cells. In the distal colon, however, fewer ketone bodies are produced, which may suggest that SCFA enter predominantly the tricarboxylic acid (TCA) cycle following oxidation. In parallel, glucose and glutamine oxidation are more relevant for energy production in small intestine and proximal colon enterocytes [129]. Recently, it has been proposed that intestinal gluconeogenesis is crucial for metabolic health, by adjusting which SCFA (butyrate via c-AMP, or propionate via FFAR3/GPCR41) the colonocyte will utilize for energy production [410,411]. In vitro studies found that cultures with propionate increased the expression of FoxP3 and IL-10, leading to colonic Treg proliferation via GPCR43, also known as FFAR2 [55,56], also emphasizing its role for the immune system and potential anti-inflammatory aspects. In human adults at increased risk of colorectal cancer, a dietary intervention with green leafy vegetables reduced oxidative stress and inflammatory markers such as TNFα [354]. As for MCTs/SMCTs, SNPs in GPCRs can result in decreased potency of SCFA action [99]. Similarly, FFAR2 rs416633 was reported to decrease monocyte percentage and increase neutrophil counts in the European population [276], further supporting the role of polygenic risk associated to complex traits regarding SCFA.

#### 4.4.2. Liver and Adipose Tissue

Studies performed in human victims of sudden death have shown that butyrate ratios decrease from 20% of total SCFA in the gut lumen, to 8% in portal blood, revealing substantial epithelial uptake and usage, with a clearance rate by the colonic epithelium of approximately 65% [130]. SCFA are then taken up by the portal circulation and further used as energy substrate by hepatocytes, particularly propionate [366]. MCTs and SMCTs are also the means of SCFA uptake in the liver. In cells that use lactic acid as a substrate for lipogenesis and gluconeogenesis, specifically the liver, kidney tubules and adipose tissue, MCT1 and MCT2 are primarily expressed [391]. Certain authors underlined that this loop will metabolize a majority of the absorbed SCFA. Indeed, only acetate is found in measurable amounts in the systemic circulation (reaching 200 μM in venous serum) [100], while butyrate and propionate show only vestigial concentrations [6,171,412]. In a human study with 22 participants, portal concentrations of acetate, propionate and butyrate were 263, 30.3 and 30.1 mmol/l, respectively. Arterial concentrations were 173, 3.6 and 7.5 mmol/l, for acetate, propionate and butyrate respectively. Consequently, the hepatic clearance of SCFA was 4.2%, 9.8% and 5.1%, for acetate, propionate and butyrate respectively. The authors performed a sub-group analysis to observe the impact of BMI (above and below 25) and of colon resection in the production and utilization of SCFA, with no significant differences being found [65].

Den Besten analysed the metabolism of SCFA by cecal infusion of stable isotope labelled SCFA in mice. In the liver, propionate appears to be gluconeogenic after its conversion to succinate (62% used for whole body glucose production), while acetate and butyrate are rather used for fatty acid and cholesterol synthesis. Low to absent contribution from propionate to palmitate or cholesterol formation was noticed [348]. Daily propionate production from DF is estimated to be 29.5 mg/kg/day [50] for an individual of about 85 kg, and presumably makes a small contribution to endogenous glucose production [366,413]. An inulin-propionate ester given orally induced appetite reduction through peptide YY (PYY) and GLP-1 mediated mechanisms in adults with overweight, leading to weight loss and reduced intrahepatocellular lipid contents [414]. Indeed, acetate has been linked to suppression of lipolysis in adipose tissue, thus reducing free fatty acid flux to the liver and mitigating fatty liver in humans [415]. Dietary SCFA supplementation reduced obesity and insulin resistance in animal models [152], which occurred via the down-regulation of peroxisome proliferator activated receptor-γ (PPARγ) [416] in adipose tissue, which has a distinct and complementary role to hepatic PPARγ, which promotes a shift to lipid oxidation and increased energy expenditure. Interestingly, in a previous study, it was found that adipogenesis was stimulated in differentiating adipocytes through PPARγ2 up-regulation responding to acetate and propionate concentrations in a high vs low fat diet, up-regulating its receptor FFAR2, thus leading to adipose tissue accumulation [145]. Concomitantly, SCFA stimulated leptin expression via FFAR2.

However, the potential role of SCFA as signalling molecules regulating hepatic glucose homeostasis has not been fully elucidated in humans. SCFA appear to differentially regulate hepatic lipid and glucose homeostasis in an AMPK-dependent manner, involving PPAR regulated effects on gluconeogenesis and lipogenesis [416], as found for atherosclerosis, steatosis and adiposity [417], as well as for the development of T2D [418].

#### 4.4.3. Systemic Metabolism

The effects of SCFA are complex, diverse, sometimes indirect, and likely synergistic. Both acetyl-CoA and pyruvate, close relatives of SCFA, are continuously feeding the TCA cycle demonstrating the capacity of conversion and reconversion between SCFA, and the ultimate importance of acetate in energy production. Acetate can quickly be converted to acetyl-CoA and enter the TCA, increasing citrate concentrations. Propionate controls the TCA though its conversion to succinate. Butyrate is first β-oxidised and then enters the TCA as acetate. The impact of SCFA on the TCA and energy production may be further regulated depending on the receptors present on the cell membranes, as MCTs regulate influx and efflux of SCFA, as well as lactate, pyruvate and ketone bodies. Lactic acid and ketone bodies are important respiratory substrates for tissues such as the myocardium or red skeletal muscle (primarily mediated by MCT1) or the brain (mediated by MCT2 in neurons, and MCT4 in astrocytes). In cells that rely on aerobic glycolysis, such as lymphocytes, astrocytes, tumour cells and white muscle fibres, MCT4 is more abundantly expressed than MCT1. As such, MCTs shuttle lactate from glycolysing cells to respiratory cells. An example is the production of lactic acid by astrocytes, exported by MCT4 or MCT1, and taken up by nearby neurons via MCT2 or MCT1. Probably due to MCT4 having a very high affinity for pyruvate, this may reflect the need of converting pyruvate into lactate in glycolysis, in order to regenerate cytoplasmic NADH from NAD+. MCT4 is upregulated in hypoxic conditions, where an increase in intracellular lactic acid is expected. Increased concentrations of intracellular lactic acid may slow glycolysis and lead to muscular fatigue [391]. Therefore, concentrations of SCFA as well as ketone bodies and lactate can regulate nutrient access to different cell tissues synchronously. In rats, this mechanism was shown to impact long term hippocampal function, with loss of memory in case of knockdown expression of MCTs. SNPs affecting MCTs can therefore have substantial impacts in disease development in a tissue-specific manner, although human relevance and therapeutic potential of SCFA are currently unknown [75].

A recent study combining genomic, metagenomic and metabolomic analysis showed that plasma levels of acetate, rather than faecal levels of SCFA, were related to inflammatory markers (IL-10, IL-6 IL-12p70, IL-18bp) and lipid subclasses (such as VLDL-C and LDL-C), and metabolic risk score [419], given that 95% of SCFA are absorbed by colonocytes. Similarly, a recent study assessed the association between faecal and circulating levels of SCFA and insulin sensitivity in human individuals. A large variability in circulating SCFA was noticed (acetate 2.8–429.4 µmol/l, propionate 0.06–12.0 µmol/l, butyrate 0.07–6.7 µmol/l), but only circulating propionate could be predicted from faecal propionate concentrations. The study proposed that circulating, but not faecal SCFA, were associated with levels of fasting GLP-1 and lipid metabolites (acetate with fasting glycerol, propionate with fasting TAG, and butyrate with free fatty acid concentrations). Circulating acetate negatively associated with insulin sensitivity, while propionate was positively associated with insulin sensitivity in peripheral tissues. Regarding inflammatory parameters, the study did not find an association between serum SCFA and fasting PYY, IL-6, IL-8 and TNFα [420].

Indeed, following hepatic metabolism, it is estimated that plasma concentrations of acetate reach 100–150 μmol/L, propionate 4–5 μmol/L, and butyrate 1–3 μmol/L [100]. In vivo effects of SCFA are the sum of direct and indirect effects; these are dose-dependent and vary between different SCFA [66]. In order to assess the effects of SCFA on peripheral cells, SCFA concentration measures of arterial blood should be the preferred method. Unfortunately, to our knowledge, studies on arterial concentration of SCFA are scarce [64,65]. Although only significant concentrations of acetate reach terminal organs (brain, lungs, heart, pancreas), studies performed with blood-perfused liver and heart show that these organs buffer blood acetate, with uptakes above a blood concentration of 0.25 mmol and a net release below it. Thus, blood acetate is of little value as an indicator of total SCFA circulating in plasma [50]. Likewise, ratios of SCFA appear to vary substantially. In one study, a portal acetate:propionate:butyrate ratio has been described to be 58:26:16 [421], whereas others reported a ratio of 78:15:7 [422]. However, the ratio between serum propionate and serum acetate may be the best determinant of the contribution of microbial-derived SCFA to energy homeostasis in the host [348].

Regarding propionate, evidence is increasing on the important role for whole-body energy homeostasis [142,423]. Aside from gluconeogenesis in the liver, propionate stimulates intestinal lipolysis, and induces the release of GLP-1 and PYY, reducing food intake. The route of administration of SCFA may, however, have different effects in vivo as described in a recent review [424]. While investigating the effects of SCFA in a mice model of influenza infection, high fibre consumers displayed increased serum levels of all SCFA, had reduced neutrophil-induced damage to lung tissue, and both butyrate and propionate reduced pro-inflammatory molecules in the lung [425]. In this study, high fibre consumers had increases in acetate (1.82-fold over control), propionate (1.39-fold over control) and butyrate (138.5-fold over control group) [425].

Acetate is the SCFA present at highest concentrations in arterial blood. It is estimated that 0–171 μmol of acetate reach the brain, crossing the blood-brain-barrier (a rather high concentration of acetate when compared to blood concentrations), as well as 0–6 μmol of propionate (18.8 pmol/mg) and 0–2.8 μmol of butyrate (17 pmol/mg) due to different expression of GPCR41 and GPCR43 at the blood-brain-barrier.

Increased acetate production associated with dysbiosis in a high fat diet mice model promoted insulin secretion via a gut-brain-pancreas axis using the parasympathetic nervous system, resulting in increased gastrin plasma levels. Chronically increased acetate turnover appeared to induce metabolic syndrome, associated with hyperinsulinemia, insulin resistance, increased triglyceride levels and over-expression of ghrelin. Both acute and chronic effects of acetate were significantly diminished in vagotomised rats [6]. This axis was activated by acetate, but not by butyrate, which may be a negative consequence of too much acetate production, without the counterbalance of propionate at the intestinal and hepatic level. This would represent another link between dysbiosis and T2D development. Furthermore, acetate levels were significantly higher in individuals with T2D and obesity than in obese normoglycemic and healthy subjects. This study found significant correlations between HbA1c, glucose, and acetate levels, but not between acetate and C-peptide or insulin [426]. In line with positive effects of fiber, in women with T2D who received oligofructose-enriched inulin showed a significant decrease in the levels of fasting plasma glucose, HbA1c, IL-6, TNF- α and plasma LPS, as compared with maltodextrin. Decreases in levels of interferon-γ and CRP as well as an increase in the level of IL-10 were not significant between the oligofructose-enriched inulin group and the maltodextrin group [427].

### 4.5. Signalling Pathways of Interest

SCFA interact with several G-protein coupled receptors (GPCRs), particularly in the colon, skeletal muscle, liver, adipose tissue, lymphocytes and cells of the nervous system (Figure 3). Here, the relevant effect depends on the cell population, as SCFA are merely the activator of intracellular cascades. Activation of GPCR in the enteroendocrine cells lead to increased secretion of GLP-1 and PYY [428], whereas the brain-mediated activation of pancreatic β-cells leads to an increased insulin secretion [6]. GPCR identification is ongoing, with pseudogenes being currently reclassified as novel receptors/encoding genes [279].

Signalling using pattern recognition receptors (PRR), JAK/Stat, CXCR4, chemokines, inositol triphosphate and acylcarnitine shuttles may all be involved in SCFA-driven immunometabolism, which may have effects in overall homeostasis. For example, leptin, a hormone structurally belonging to the cytokine superfamily and which can activate monocytes, neutrophils and macrophages, also regulates appetite and body weight and affects basal metabolism by regulating insulin secretion. GWAS have found that SNPs of genes involved in the leptin pathway were the greatest influencing factors of microbiota colonization in the nose, oral cavity and skin [240], possibly due to modulation of mucin expression [429]. While rs7799039 and rs1137101 in leptin (*LEP*) and leptin receptor (*LEPR*) genes, respectively, did not alter circulating leptin levels, these are associated with cardiovascular disease and metabolic syndrome, with predisposed individuals presenting with increased glycated haemoglobin, insulin and increased fat mass, among other clinical phenomes [280]. Leptin may be further associated with wound healing [430] and psoriasis [431,432]. Butyrate and propionate also promote wound healing, by stimulation of epithelial migration and differentiation through p21 activated kinase (PAK1) and milk fat globule-EGF factor 8 (MFGE8) [433].

Plasma leptin concentration is negatively correlated to Aβ levels in Alzheimer’s disease (AD). Indeed, AD animal models of AD treated with leptin showed a reduction in Aβ and phosphorylated tau levels. SCFAs have therefore indirect but also direct possible therapeutic potential in neurodegenerative diseases. Firstly, SCFA act as substrates for the synthesis of serotonin. Further, butyrate acts as a HDAC inhibitor capable of restoring fear learning, counteracting intraneuronal Aβ deposition, and butyrate, valerate and propionate have attenuated AD progression by inhibiting Aβ oligomerization [14].

## 5. Conclusions and Perspectives

### 5.1. Main Conclusions

Currently, it is estimated that the DF intake in Europe for adult males is around 18 to 24 g/d and for females 16 to 20 g/d, with little variation between countries, falling below most national AI recommendations, i.e., an intake of 25–35 g for adults (25–32 g/d for adult women and 30–35 g/d for adult men) [40]. Sufficient intake of DF alone would contribute to an estimated 15–30% reduction in NCDs [107]. Overall, SCFA produced from dietary fibres may exert profound systemic effects, although they are strongly associated with secondary plant metabolites, which are likely concomitantly taken up by the host, and may further contribute to observed health benefits.

SCFA are the major carbon flux shared between the GM and the host, and regulatory roles in local and peripheral metabolism are emerging [366]. Many of these aspects are potentially related to the effect of SCFA on immune and inflammatory modulation pathways. It is not unreasonable to wonder whether SCFA represent a key molecular link between diet, the gut microbiome and health [366]. However, the causality of microbiota-derived metabolites in the aetiology of human disease remains unclear. Butyrate has received much attention due to its effect on cell proliferation studied in vitro, but more research is warranted in order to understand the roles of non-butyrate SCFA, i.e., propionate, acetate, valerate and caproate.

Such research should be based preferably on human studies. Several authors have warned about the implications of extrapolating in vitro (studies often performed in cancer or immortalized cells) findings of SCFA, i.e., anti-inflammatory, anti-cancer or epigenetic effects such as HDAC inhibition, to humans [100,434], as in vitro cell cultures may use other than normal metabolic pathways. Indeed, butyrate and propionate concentrations are higher in cancer cells due to the Warburg effect, leading to reduced genetic transcription than could be expected in healthy cells in vivo [100]. However, such mechanistic insights provide knowledge on the association of genomes, disease phenotypes and microbial taxa [238].

The enterotype, i.e., the clustering of gut bacterial communities, represents the first level of inter-individual variability in vivo. A changed enterotype may result in substantial gastrointestinal and systemic impact. In vivo effects of SCFA are the sum of direct, and indirect effects; these are not only dose-dependent, but vary between different SCFA [66]. Colonocytes take up SCFA and derive a large extent of their energy requirement from β-oxidation of SCFA. Non-metabolized SCFA enter the bloodstream and are metabolized in the liver, skeletal muscle and brain, among others. As the effect of SCFA is pleiotropic, eliciting intracellular cascades can be lipogenic or gluconeogenic, tolerogenic or immunogenic [62,98,156,375,435]. Butyrate appears to impinge on enterocyte proliferation in a dose-dependent manner, inducing both cell proliferation in healthy intestinal crypts, as well as apoptosis in metaplastic cells, via p53 [404]. Acetate and propionate, thought to be tolerogenic, are rather decreased (though not significantly) in a fibre-free diet, although butyrate levels in faeces remain stable [100].

Slow intraluminal SCFA diffusion rate, rapid mucosal absorption (>95%) and enterocyte metabolism of SCFA, are some of the factors in vivo that cause the estimation of SCFA production from faecal samples rather imprecise [129,436]. This measurement only allows drawing conclusions based on the approximately 5% of SCFA that remain unabsorbed following their colonic passage. Furthermore, luminal concentrations of SCFA measured in animals were criticized for not reflecting purely their rate of production, but instead their rate of epithelial absorption, which is even further modulated by luminal lactate concentrations and pH [353]. With regard to circulating concentrations, blood acetate seems to be of little value as an indicator of total SCFA circulating in plasma. The ratio between serum propionate and serum acetate may be the best determinant of the SCFA microbial-contribution to whole-organism homeostasis [348].

SCFA production has clearly been shown to be related to the amount of DF intake, as well as to the composition of the GM. While the majority of GM are obligate anaerobes [437,438], making their study difficult with traditional methods, metagenomic approaches may allow the complete taxonomy of the GM soon to be known. However, understanding the metabolic complexity of the holobiont will undoubtedly require more time and efforts. Depending on the host’s GM, the degree of DF fermentation and therefore SCFA production and their uptake will vary. *Bacteroidetes* spp. mainly produce acetate and propionate, *Prevotella* is an acetate producer, and the *Firmicutes* phylum tend to produce butyrate [183]. Many cross-feeding interactions exist between these phyla, which are the most abundant in the human gut. Given that the GM composition has not been found to have characteristic changes in relation to specific phenotypes, we can hypothesize that some genotypes will benefit from one or the other enterotype for improved health status. Nevertheless, in vivo studies of total DF consumption, GM composition, circulating concentrations of SCFA and genetic analysis in healthy humans are scarce and present heterogeneous results, possibly due to study design or the use of various types of dietary fiber [182].

The second level of variability is the individual’s host genome. Recent studies have found significant correlations between different cellular signalling pathways and specific microbial colonization in individuals [240]. The feedback between the GM and the genome is reflected in the regulation of the mucus layer (responding to DF intake), potency of enzymatic activity (as seen for *AMY1* and *LCT*), or reactivity of receptors and transporters present in colonic strata, such as SMCTs and MCTs. SNPs in genes such as *AMY1*, *MUC2*, *FUT2*, SCFA receptors (*MCT1-4*, *GPCR41, GPCR43, GPCR109A*) and tight junction proteins, to name a few, may strongly modulate the gut microbiota composition [3,61,251]. Acetate, propionate and butyrate are taken up by specific MCTs in the gut epithelium and are widely distributed in human tissues. Furthermore, both SCFA and their receptors GCPR109A, GPCR41 and GPCR43 have been previously associated with specific disease phenotypes such as metabolic syndrome, Parkinson disease, cancer, gastrointestinal disorders and T2D [6,35,202,439,440]. However, authors note that much larger sample sizes are needed to elucidate the remaining effects of host genetics on the gut microbiome [359].

The (combined host-microbe) metabolic steps of DF reveal the deep symbiosis existing between the human host and their microbiome. In this context, DF and the mucus barrier in the colon, seem to be the strongest mechanism linking both genomes [192]. According to the holobiont theory, host and microbe genomes, including individual genes, are selected if advantageous for the holobiont. This implies that the microbiome and the host are attempting to achieve homeostasis through cooperative mechanisms. Blekhman [240] and Bonder [441], among others, have emphasized the importance of genetic variation and associated microbiota, with innate immunity genes being highly conserved and correlated with microbial taxa.

The third level of variability, the phenotypical level, refers to the resultant interplay of the genome, the microbiome and lifestyle factors, of which diet is thought to be a significant contributor [44,442]. The mentioned examples of *Prevotella*, *Akkermansia*, and *E. limosum* may reflect an intricate symbiont homeostasis associated with dietary patterns, resulting in observable phenotype trajectories. DF and produced SCFA may influence all phenotypes to a certain degree, whether by maintaining homeostatic balance [6], by modulating the GM via substrate competition [442], by decreasing the rate of disease progression as a result of reduced mean arterial pressure and heart rate [412], by reducing disease symptoms due to anti-inflammatory actions [51], or by modulating pharmacological responses, potentially by interacting with the further transport of drugs [240,286,379]. However, the impact of various DF-associated compounds acting as transport inhibitors must also be taken into account [389], as the same receptors are used for drug delivery. These and other microbial derived metabolites represent a new frontier in understanding pathogenesis and physiology [366].

Understanding the inter-individual differences regarding the effect of DF, including the metabolism of SCFA and potential health outcomes, is complex. In specific organs, a balance between aerobic and anaerobic cells is achieved, with SCFA at its core. Some studies further delegate SCFA a dominant role in cell differentiation, as seen for macrophages and colonocytes. Indeed, as close relatives of members of the TCA or the mitochondrial respiration chain, SCFA concentrations may strongly impact these energy systems, both at the cellular level, as well as at the systemic level (e.g., insulin and glucose control) [142,426,427].

Thus, microbial metabolites including SCFA may be suitable biomarkers in clinical practice, as their detection in blood plasma and possibly also in faeces is usually affordable, and may precede the onset of clinically manifest disease symptoms. Furthermore, naturally occurring DF is also covalently bound to other phytochemicals, such as polyphenols [44], and other compounds such as carotenoids can be entrapped [44]. These secondary plant metabolites would therefore reach the colon, where they may further exert direct and indirect antioxidant and anti-inflammatory effects by, e.g., its redox potential, upregulation of Nrf2 or downregulating NF-κB respectively, and may contribute to the positive health effects of DF [44,341]. A number of health effects may be attributed to DF-bound phytochemicals, with or without microbial bioconversion, presenting with potential interferences on the hormonal level, such as lignans [433]. However, little is known on factors explaining inter-individual differences in response to DF-bound phytochemicals, and their levels may vary drastically, depending on the type of DF consumed.

### 5.2. Future Perspectives

Studies are consensual in demonstrating that the GM can shift dramatically upon dietary changes, but also that it is quite resilient and will quickly return to a stable baseline [88]. Precision dietary interventions, faecal microbiota transplantation and complementary or synergistic synbiotics (formulations of pre- and probiotics) are all technologies undergoing rapid evolution. These are expected to have a beneficial impact on a broad range of diseases, from paediatric diseases such as autism spectrum disorders or atopy and allergy, to age-associated conditions such as neurodegeneration and cancer, as well as food intolerances [197].

Some authors have observed that the increased prevalence of NCDs in the last fifty years may not be related to “losing microbes” or “proliferation of individual pathogens”, but instead to the evolution of environmental exposures, including the increased consumption of ultra-processed food and decreased DF consumption [196]. These authors suggested a relationship between NCDs and a different interplay between our “new” microbiota with our “old” genes; i.e., as human diet has shifted in the last 50 years, e.g., to more sugar, more saturated fatty acids and less fibre intake, so has the microbiome. The authors postulate that the low-grade chronic pro-inflammatory status associated with NCDs is the result of dietary changes and incomplete adaptation by the holobiont, including the microbiota [443].

Highly complex interactions involving the human host, GM and dietary patterns result in the creation of large databases with numerous variables, many of which, e.g., immunity or human genetics, may not be yet fully characterized [444]. Wolter et al. pointed out that future research must have as an aim to identify both general and subpopulation-specific biomarkers, in order to understand the underlying mechanisms behind varied responses to standardized interventions [444]. While microbiota-focused treatment options, such as through DF and modulation of SCFA production may become highly relevant in the future, consensual “core optimal” or “healthy” microbiome needs further defining, in a highly individualized and self-regulated system. This has led to several authors affirming that translational research—in an attempt to complement fundamental research, animal models and mechanistic studies with epidemiological studies, human intervention studies, deep phenotyping and longitudinal study designs—may be the necessary next step to gain insight into the nutritional interactions taking place in vivo, in order to explain DF variations among individuals [445,446].

Although investigating the associations between dietary patterns, GM and host genetics is a promising field of study, the specific mechanisms providing phenotypical differences and specific disease trajectories over lifetime remain unknown. Deep phenotyping and adapted study designs, such as N-of-1 methods performed in the frame of randomised controlled trials (RCTs) may be required for such insights [162]. Longitudinal follow-up studies of dietary interventions and observation across many years may be necessary to evaluate the real polygenic risk impact of common SNPs related to SCFA metabolism in the population. In addition, longitudinal population-based studies are necessary to confirm the relationship between polygenic risk and common SNPs in the general population. Precision, patient-tailored therapies combined with measuring SCFA may be possible in the future following pharmacological or nutritional intervention, potentially without strong interference in overall homeostasis and wellbeing [238].

Enterotyping and genetic sequencing will have to be complementary to each other, as to understand the impact of such metabolites in individual persons. In future research, polygenic risk scores of common polymorphisms, using novel tools, should be scrutinized.While further research is needed before drawing any conclusions, this review elucidated the potential of SCFA as biomarkers for future healthcare, while taking into account potential factors explaining individual variability of responses.

## Figures and Tables

**Figure 1 nutrients-14-05361-f001:**
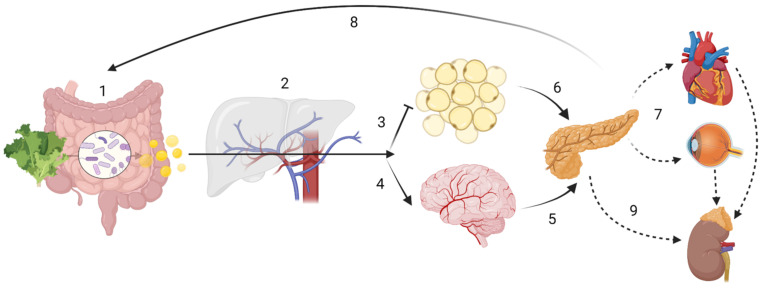
Host-driven variability in SCFA metabolism and distribution may lead to different disease outcomes. ADME (sub-) steps may explain the variability in SCFA effects. The enterotype influences the amount of SCFA produced, while human digestive enzymatic activity may regulate microbial communities; (1) Absorption: SNPs in mucin, MCTs or tight junction function could impair SCFA bioavailability. Butyrate is the main energy source for colonocytes. (2) In the portal circulation SCFA undergo first-pass effects, where a majority of propionate is metabolized via GPR109A, GPR43 and GPR41, having gluconeogenic or lipogenic effects. Distribution In the systemic circulation: although at present at low concentrations, butyrate and propionate are still detectable; acetate is now the most abundant SCFA. (3) Acetate inhibits lipolysis at the adipose tissue level. (4) Acetate can cross the “blood-brain-barrier” (BBB). Metabolism: SCFA have showed to be effective against microglial oxidative stress responses. SCFA may also have cellular signalling properties, as evidenced by its control of centrally released insulin (6) or its impact on the hypothalamic-pituitary-adrenal axis in leptin and cortisol responses, which may ultimately lead into maladaptive health conditions across the body (7). Finally, gluconeogenic, lipogenic and insulinogenic signals impact ghrelin, leptin and peptide YY release, leading to appetite suppression and satiety (8), improved insulin sensitivity and glucose metabolism, as well as reduction of serum lipids. (9) Excretion: in the kidney, SCFA can be re-absorbed by MCT1. Note: the intracellular effect of SCFA e.g., on HDAC or NF-κB are not displayed. Created with BioRender.com.

**Figure 2 nutrients-14-05361-f002:**
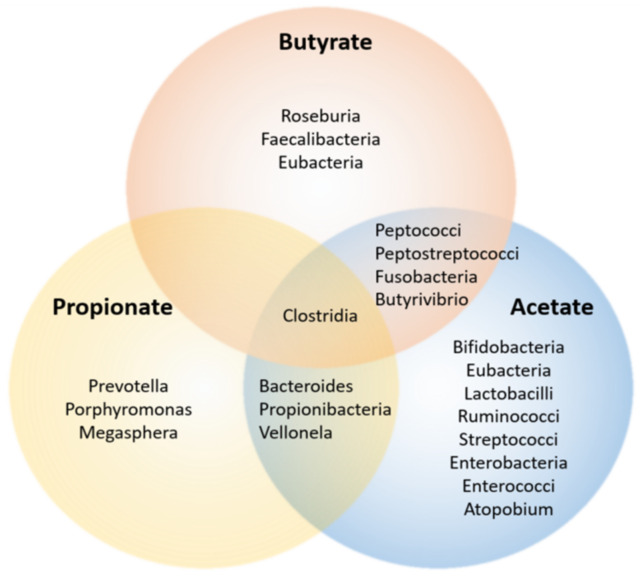
SCFA-producing microbiota. Different bacterial taxa are associated with the production of different SCFA. Of note, the Clostridium family is not associated with a particular SCFA. This may reflect the abundance of different species of the Clostridia genus in the human gut. Adapted from Macfarlane and Macfarlane [129].

**Figure 3 nutrients-14-05361-f003:**
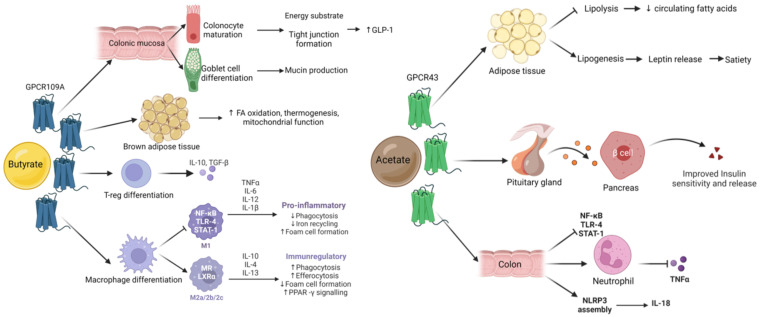
Pleotropism of SCFA, acetate and butyrate. Each SCFA seem to have some organ specificity; butyrate is mainly used for energy generation at the colonic level. In the liver, propionate is principally metabolized, where pleotropic action is found, being lipogenic or gluconeogenic based on its concentration. Acetate is found to affect the hypophysis-adrenal gland axis. However, the same SCFA may impact different organs through different receptors, as receptors show preference but are not restricted to a single SCFA, possibly leading to synergistic effects. Created with BioRender.com.

## Data Availability

Not applicable.

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
