# Peer review of "Short Chain Fatty Acid Metabolism in Relation to Gut Microbiota and Genetic Variability"

_nutrients, 2022, doi:10.3390/nu14245361_

Round 1

Reviewer 1 Report

The authors have done a commendable job with this review. It is extensive and very nicely structured with nice and useful tables and figures. I found the reading fluid and easily understandable. My only concern is that recent studies have shown gut microbiota-mediated regulation of metabolism also has a 'circadian' component. Indeed, microbiota regulates intestinal clock, and in turn intestinal microbiota display diurnal rhythmicity. Furthermore, majority of metabolism display diurnal rhythmicity. In this sense, if authors could introduce a small subsection to talk about circadian rhythm and microbiota, the review will be fantastic. This subsection need not be large.

Author Response

The authors have done a commendable job with this review. It is extensive and very nicely structured with nice and useful tables and figures. I found the reading fluid and easily understandable. My only concern is that recent studies have shown gut microbiota-mediated regulation of metabolism also has a 'circadian' component. Indeed, microbiota regulates intestinal clock, and in turn intestinal microbiota display diurnal rhythmicity. Furthermore, majority of metabolism display diurnal rhythmicity. In this sense, if authors could introduce a small subsection to talk about circadian rhythm and microbiota, the review will be fantastic. This subsection need not be large.

Reply: Thank you for your appraisal of the review article, and for your pertinent suggestion. Although we were aware of this rhythmical component of the gut microbiota, we have indeed failed to include it earlier. Indeed, diet will influence both host and microbiota metabolism, and this is a growing  area of interest, also as a therapeutic aspect of complex diseases such as diabetes, cardiovascular disease and even psychological well-being. More research is needed in this field, therefore this was now added in lines 522 ff.

Reviewer 2 Report

Interesting review which provides comprehensive information related to short chain fatty acid. It would be nice if the author elaborate a bit more about the dietary fiber category source (with latest reference), and health benefits. The importance of lactobacillus/bifidobacterium against toxin-induced systemic inflammation needs to be expanded with recent citations.

Author Response

Interesting review which provides comprehensive information related to short chain fatty acid. It would be nice if the author elaborate a bit more about the dietary fiber category source (with latest reference), and health benefits. The importance of lactobacillus/bifidobacterium against toxin-induced systemic inflammation needs to be expanded with recent citations.

Reply: Thank you for your appraisal of the review article, and for your pertinent suggestions. Regarding dietary fibre health benefits, we provided recent articles pertaining to several nontransmissible chronic diseases, such as CVD, depression, cognitive decline, multiple sclerosis, Parkinson’s disease, osteoarthritis and gastrointestinal conditions like irritable bowel syndrome (IBD) in lines 56ff.

We further added examples of C. difficile infection (not an NCD) and paediatric kidney disease,  in lines 61 ff. (highlighted).

For dietary fibre category sources, we now have detailed this further, although this tends to be self-reported data with wide variation across the globe, in lines 130 ff.

We have also added an interesting secondary analysis of the DIETFITS trial, which clarifies main fiber sources in low-fat vs low-carbohydrate diets in USA, now further expanded in lines 148 ff.

The importance of Lactobacillus and Bifidobacterium in toxin-induced systemic inflammation in either reducing LPS activation via TLR-4 of intestinal macrophages or its translocation to the circulation by reducing gut leakiness, was added now in lines 599 ff.